# Photodissociation of particulate nitrate as a source of daytime tropospheric Cl$_2$

Xiang Peng [1,2], Tao Wang [1✉], Weihao Wang[1,3], A. R. Ravishankara [4], Christian George [5], Men Xia[1], Min Cai[6], Qinyi Li [7], Christian Mark Salvador[8,15], Chiho Lau[9], Xiaopu Lyu[1], Chun Nan Poon[1], Abdelwahid Mellouki [6], Yujing Mu [10], Mattias Hallquist [8], Alfonso Saiz-Lopez [7], Hai Guo[1], Hartmut Herrmann [11,12], Chuan Yu[1,13], Jianing Dai[1,16], Yanan Wang[1], Xinke Wang[5], Alfred Yu[9], Kenneth Leung[9], Shuncheng Lee[1] & Jianmin Chen [14]

Chlorine atoms (Cl) are highly reactive and can strongly influence the abundances of climate and air quality-relevant trace gases. Despite extensive research on molecular chlorine (Cl$_2$), a Cl precursor, in the polar atmosphere, its sources in other regions are still poorly understood. Here we report the daytime Cl$_2$ concentrations of up to 1 ppbv observed in a coastal area of Hong Kong, revealing a large daytime source of Cl$_2$ (2.7 pptv s$^{-1}$ at noon). Field and laboratory experiments indicate that photodissociation of particulate nitrate by sunlight under acidic conditions (pH < 3.0) can activate chloride and account for the observed daytime Cl$_2$ production. The high Cl$_2$ concentrations significantly increased atmospheric oxidation. Given the ubiquitous existence of chloride, nitrate, and acidic aerosols, we propose that nitrate photolysis is a significant daytime chlorine source globally. This so far unaccounted for source of chlorine can have substantial impacts on atmospheric chemistry.

[1] Department of Civil and Environmental Engineering, the Hong Kong Polytechnic University, Hong Kong 999077, China. [2] Department of Ambient Air Quality Monitoring, China National Environmental Monitoring Center, Beijing 100012, China. [3] Hangzhou PuYu Technology Development Co., Ltd, Hangzhou, Zhejiang 311300, China. [4] Departments of Atmospheric Science and Chemistry, Colorado State University, Fort Collins, CO 80523, USA. [5] Univ Lyon, Université Claude Bernard Lyon 1, CNRS, IRCELYON, Villeurbanne 69626, France. [6] Institut de Combustion, Aérothermique, Réactivité et Environnement (ICARE), CNRS/OSUC, 45071 Orléans, Cedex 2, France. [7] Department of Atmospheric Chemistry and Climate, Institute of Physical Chemistry Rocasolano, CSIC, Madrid 28006, Spain. [8] Department of Chemistry and Molecular Biology, University of Gothenburg, Gothenburg 40530, Sweden. [9] Air Science Group Environmental Protection Department, HKSAR, Hong Kong 999077, China. [10] Research Center for Eco-Environmental Sciences, Chinese Academy of Sciences, Beijing 100085, China. [11] Leibniz Institute for Tropospheric Research (TROPOS), Atmospheric Chemistry Department (ACD), 04318 Leipzig, Germany. [12] School of Environmental Science and Engineering, Shandong University, Qingdao, Shandong 266237, China. [13] Environment Research Institute, Shandong University, Qingdao, Shandong 266237, China. [14] Department of Environmental Science and Engineering, Fudan University, Institute of Atmospheric Sciences, Shanghai 200433, China. [15] Present address: Balik Scientist Program, Department of Science and Technology - Philippine Council for Industry, Energy and Emerging Technology Research and Development, Bicutan, Taguig 1630, Philippines. [16] Present address: Environmental Modeling Group, Max Planck Institute for Meteorology, Hamburg 20146, Germany. ✉email: cetwang@polyu.edu.hk

Atomic chlorine (Cl) is a very reactive radical, known to destroy stratospheric ozone ($O_3$) through catalytic cycles[1,2]. In the lower troposphere, it can initiate the oxidation of volatile organic compounds (VOCs), increase the levels of conventional radicals (OH, $HO_2$ and $RO_2$), and produce $O_3$ and secondary organic aerosols (SOA)[3–7] which are air pollutants and also alter the Earth's radiation budget and climate. Cl reacts rapidly with methane, the most abundant hydrocarbon and the second-most important greenhouse gas emitted into the atmosphere[8,9]. Molecular chlorine ($Cl_2$) is an important Cl precursor. It can be photolyzed quickly to release two Cl atoms during the daytime, and its production through heterogeneous reactions is a key step in the $O_3$ destruction over Antarctica during austral spring[10]. Previously, $Cl_2$ has been measured in the lower troposphere in locations such as at the Arctic surface[11,12], the marine boundary layer[13–15], and continental sites[16,17]. $Cl_2$ was found to typically peak during nighttime, but elevated levels (17–450 pptv) have also been observed during daytime[6,11,12,18–21]. The daytime occurrence of $Cl_2$ is of great importance as it may have a profound impact on atmospheric photochemistry and oxidation capacity[6,19]. Such observations also reveal the existence of a significant $Cl_2$ source that compensates or even overcomes its fast photolytic loss. Although daytime $Cl_2$ can be emitted from various sources, such as from coal combustion[16] or water treatment facilities[15], it can also be produced through some photochemical processes[11,18–20]. However, the underlying photochemical mechanisms remain uncertain. As a result, current state-of-the-art air quality models do not typically implement such chemistry, and therefore, cannot reproduce the observed high daytime $Cl_2$ levels in polluted regions[7,22]. Consequently, the impact of $Cl_2$ on atmospheric oxidation is currently underestimated. Finally, as there were just only handfuls of $Cl_2$ observations outside the polar regions to date[13–17,19–21], our ability to assess the $Cl_2$ and Cl impact in different parts of the world is still very limited.

In this work, we report atmospheric observations of $Cl_2$ and other chemicals obtained at a polluted coastal site in southern China during autumn 2018. The $Cl_2$ concentrations are much higher than those previously measured outside polar regions. We show that previously proposed $Cl_2$ production mechanisms cannot account for the large $Cl_2$ daytime source at this site and this source positively correlates with solar radiation, particulate nitrate, and particulate surface area. Laboratory experiments show that illuminating solution of sodium chloride and nitrate under acidic condition and ambient particulates can produce a large amount of $Cl_2$, which can explain a large fraction of the observed $Cl_2$ at our site. We propose that nitrate photolysis at high aerosol acidity is an important pathway for activating inert chloride to produce photoliable $Cl_2$ during daytime in the polluted troposphere. Model calculations demonstrate significant enhancement of conventional radical levels, hydrocarbon oxidation, and ozone production by the high levels of $Cl_2$ at the study site. We suggest that the same $Cl_2$ production pathway may exist in other places of the world and call for more attention to the role of $Cl_2$ in tropospheric chemistry and air quality of polluted regions.

## Results and Discussion

**Field observations.** To investigate the abundance, sources, and impact of $Cl_2$, we measured its concentrations using a chemical ionization mass spectrometer (CIMS) (Methods section "Field measurements") at a coastal site in Hong Kong (Cape D'Aguilar, 22.21°N, 114.25°E, Supplementary Fig. 1), adjacent to the highly industrialized Pearl River Delta (PRD). The field measurement took place from 31 August to 9 October 2018, when this site predominantly received outflow of air from eastern and southern

China and occasionally inflow of marine air and spillover of urban pollution from Hong Kong (HK) and other PRD cities[23,24] (Fig. 1A, B, also see Methods section "HYSPLIT and E-AIM models"). Moderate to very high mixing ratios of ozone (up to 186 ppbv) (Supplementary Fig. 2) were observed during the study, indicating active photochemistry during the measurement period.

We frequently observed $Cl_2$ mixing ratios greater than 400 pptv (10-min average) with a maximum of 998 pptv (Supplementary Fig. 2), which is much higher than the values reported in the limited $Cl_2$ measurements[6,9,19]. The $Cl_2$ mixing ratio exhibited a distinct daytime peak (Fig. 1C and Supplementary Fig. 2), coinciding with that of ozone. Much higher $Cl_2$ levels were observed in the air mass originating from inland than that from the ocean, indicating the important role of anthropogenic pollution in producing the observed high $Cl_2$ (Fig. 1C, D). The highest $Cl_2$ (and $O_3$) occurred on 11 September 2018 in a heavy photochemical pollution episode (Supplementary Fig. 2), when the site was impacted by plumes from HK and other PRD cities[25]. With an average photolysis lifetime of $Cl_2$ of about 7 min at noon during this study, sustained high levels of daytime $Cl_2$ must arise from a significant in-situ production, with an average production rate of up to 2.7 pptv s$^{-1}$ at noon. $ClNO_2$—another Cl precursor —exhibited typical nighttime peaks with the highest mixing ratio of 1900 pptv, comparable to the value observed in our previous measurements at a nearby site[26].

Previous studies have proposed two chemical mechanisms to explain the observed daytime $Cl_2$ production. The first one involves the aqueous-phase reaction of OH with chloride in the solution or at the air-water interface, with OH being produced from $O_3$ photolysis in the gas or aqueous phase (R1–R9)[27,28].

$$O_{3(g)} + h\nu \rightarrow O(^1D)_{(g)} + O_{2(g)} \tag{R1}$$

$$O(^1D)_{(g)} + H_2O \rightarrow 2OH_{(g)} \tag{R2}$$

$$OH_{(g)} + Cl^-_{(interface)} \rightarrow (OH\cdots Cl^-)_{(interface)} \rightarrow 1/2\, Cl_{2(g)} + OH^-_{(aq)} \tag{R3}$$

$$OH_{(aq)} + Cl^-_{(aq)} \rightleftharpoons HOCl^-_{(aq)} \tag{R4}$$

$$HOCl^-_{(aq\, or\, interface)} + H^+_{(aq)} \rightleftharpoons HClOH_{(aq)} \tag{R5}$$

$$HClOH_{(aq)} \rightleftharpoons Cl_{(aq)} + H_2O \tag{R6}$$

$$Cl_{(aq)} + Cl^-_{(aq)} \rightarrow Cl^-_{2(aq)} + H_2O \tag{R7}$$

$$2Cl^-_{2(aq)} \rightarrow Cl_{2(aq)} + 2Cl^-_{(aq)} \tag{R8}$$

$$Cl_{2(aq)} \rightarrow Cl_{2(g)} \tag{R9}$$

This mechanism was based on laboratory observations of the production of 10–100 ppbv of $Cl_2$ when gaseous $O_3$ (0.8–14 ppmv) and deliquesced sea salt particles were illuminated with 254 nm ultra-violet light[27]. The experimental results were supported by molecular dynamics and kinetics calculations[28]. These studies revealed a maximum $Cl_2$ production of 375 pptv s$^{-1}$ (with 14 ppmv $O_3$ and a photolysis rate constant for $O_3$ to generate O($^1D$) (J($O_3 \rightarrow$O($^1D$)) of $7.92 \times 10^{-4}$ s$^{-1}$). This production rate was extrapolated to typical mid-Atlantic conditions, assuming that the $Cl_2$ production was proportional to the level of ozone and solar radiation and Cl$^-$ availability was sufficient. These conditions explained the observed $Cl_2$ at a coastal site in Long Island, New York[27]. If we extrapolate their production rate, with the same assumptions, to our ambient conditions i.e., $O_3$ (65 ppbv) and J($O_3 \rightarrow$O($^1D$)) ($1.78 \times 10^{-5}$ s$^{-1}$), which is calculated from the

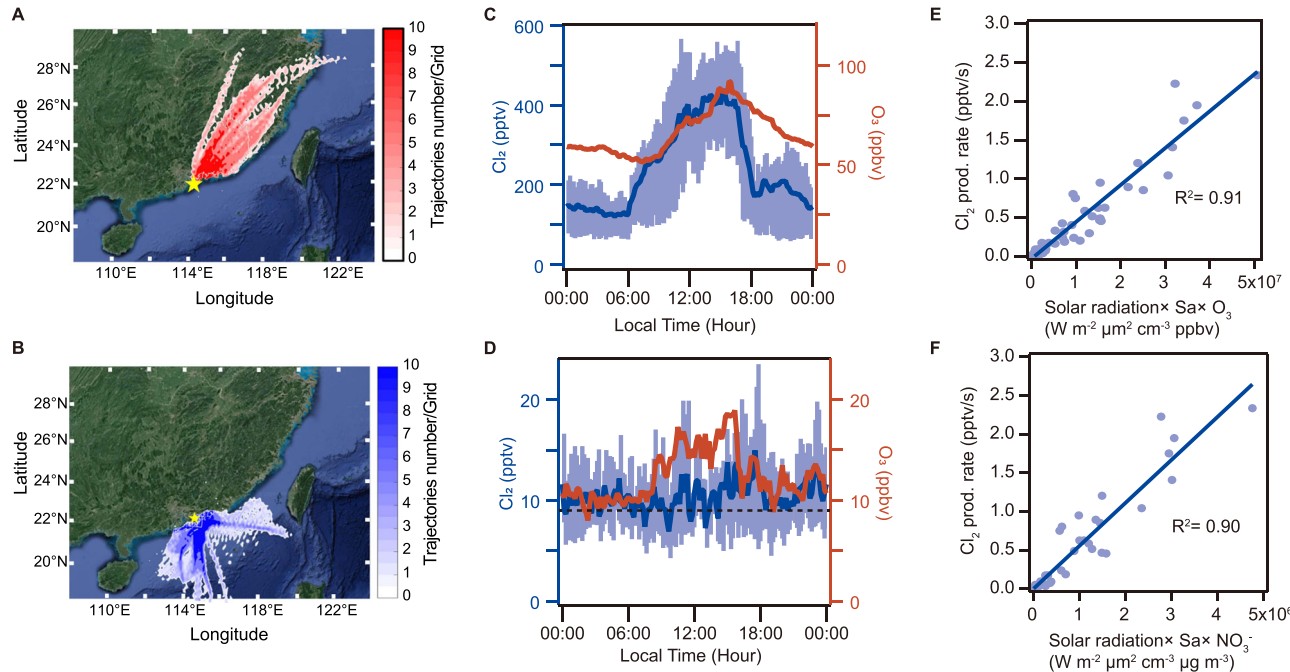

**Fig. 1 Ambient observations at Cape D' Aguilar, Hong Kong, from 31 August to 9 October 2018.** Back trajectories of air mass from (**A**) continental region (5 September - 9 October) and (**B**) the South China Sea (31 August - 4 September). Contour represents the number of trajectories in each 0.1-degree latitude × 0.1-degree longitude grid. The diurnal profiles of $Cl_2$ and $O_3$ (**C**) in the air mass from the continental region (5 September - 9 October); (**D**) in the air mass from the South China Sea (31 August - 4 September). The dashed line represents the detection limit of the CIMS instrument. The blue line is the 10-min average of $Cl_2$, and the blue shade represents the 25 percentile and 75 percentile values. The red line is the 10-min average of $O_3$. (**E**) The scatter plot of the production rate of $Cl_2$ ($P_{Cl_2}$) and the product of the solar actinic flux (W m$^{-2}$), the aerosol surface area density (Sa, μm$^2$ cm$^{-3}$), and $O_3$ mixing ratio (ppbv) from 08:00 to 18:00 in the continental air mass. (**F**) The scatter plot of the production rate of $Cl_2$ ($P_{Cl_2}$) and the product of the solar actinic flux, the aerosol surface area density (Sa, μm$^2$ cm$^{-3}$), and nitrate concentration in PM$_{10}$ (μg m$^{-3}$) from 08:00 to 18:00 in the continental air mass. The $P_{Cl_2}$ equals the photolysis rate of $Cl_2$ ($J_{Cl_2}$ × measured $Cl_2$ concentration) as $Cl_2$ was near a photo stationary state. $J_{Cl_2}$ was calculated from the TUV model (http://cprm.acom.ucar.edu/Models/TUV/Interactive_TUV) under clear sky conditions and then scaled to the solar radiation derived $J_{NO2}$ (see Methods section "Chemical box model").

TUV model under clear sky condition, the $O_3$ photolysis would produce $Cl_2$ at a rate of 0.039 pptv s$^{-1}$, which is one order of magnitude smaller than the average daytime (08:00–18:00) production rate (P($Cl_2$)) of 0.46 pptv s$^{-1}$ measured at our site. Here the P($Cl_2$) is assumed to be equal to the photolysis rate of $Cl_2$, as the $Cl_2$ is nearly in a photo-stationary state (considering its short lifetime of ~7 min at noon in our study).

Another suggested mechanism is the classic autocatalytic halogen activation, which begins with a Cl atom reacting with $O_3$ to form chlorine monoxide (ClO) during daytime (R10–R11). ClO further reacts with $HO_2$ or $NO_2$ to form hypochlorous acid (HOCl) (R12) or chlorine nitrate (ClONO$_2$) (R13), respectively. These two compounds can then undergo photolysis or react on acidic chloride-containing aerosol particles to form $Cl_2$ (R14–R15) that partitions to the gas phase[9,29].

$$Cl_{2(g)} + h\upsilon \rightarrow 2Cl_{(g)} \tag{R10}$$

$$Cl_{(g)} + O_{3(g)} \rightarrow ClO_{(g)} + O_{2(g)} \tag{R11}$$

$$ClO_{(g)} + HO_{2(g)} \rightarrow HOCl_{(g)} + O_{2(g)} \tag{R12}$$

$$ClO_{(g)} + NO_{2(g)} + M \rightarrow ClONO_{2(g)} + M \tag{R13}$$

$$HOCl_{(g)} + H^+ + Cl^- \xrightarrow{aerosol} Cl_{2(g)} + H_2O_{(aq)} \tag{R14}$$

$$ClONO_{2(g)} + H^+ + Cl^- \xrightarrow{aerosol} Cl_{2(g)} + HNO_{3(aq)} \tag{R15}$$

We used a photochemical box model[6] (also see Methods section "Chemical box model") to simulate HOCl and ClONO$_2$

(Supplementary Fig. 3) based on known gaseous chlorine chemistry, by constraining it to the observed $Cl_2$ and other chemical constituents concentrations (Supplementary Table 1 and Supplementary Fig. 4). The calculations were performed for the period 4—14 September 2018, for which a more complete VOC dataset is available. The simulated mixing ratios of HOCl were a factor of 3 lower than those of $Cl_2$ (Supplementary Fig. 3), as Cl atoms predominantly react with volatile organic compounds (VOCs) (~83%) but less efficiently with ozone (~17%) to form ClO and then HOCl at our site (see below). The calculated $Cl_2$ production rate (via R14) was two orders of magnitude lower than the observed rate, even if we adopt the highest model-predicted HOCl value (180 pptv) and previously reported the highest HOCl uptake coefficient of 0.0002 (Methods section "Estimation of Cl2 production from heterogeneous reactions of HOCl"), confirming the negligible role of HOCl in producing $Cl_2$ (via R14) at our site. For ClONO$_2$, the model calculated mixing ratios (Supplementary Fig. 3) were two orders of magnitude lower than the observed $Cl_2$ values, suggesting its insignificant role in $Cl_2$ production (via R15). To conclude, the previously two known mechanisms for producing daytime $Cl_2$ cannot account for the high $Cl_2$ production observed, and the mismatch is larger than an order of magnitude.

To gain more insight into the potential sources of daytime $Cl_2$, we examined the relationship between P($Cl_2$) and various measured parameters (see Supplementary Fig. 5) that might be involved in the $Cl_2$ photochemical production. We found a good correlation between P($Cl_2$) and the product of the solar actinic flux and the aerosol surface area density ($R^2 = 0.71$) (Supplementary Fig. 5), and the correlation was further improved with

consideration of $O_3$ ($R^2 = 0.91$) (Fig. 1C) or nitrate in aerosol ($R^2 = 0.90$) (Fig. 1D). The high correlation between the product of $O_3$ abundance and surface area density and $P(Cl_2)$ is not necessarily the result of a causal relationship between $O_3$ and $Cl_2$, but likely highlights their photochemical co-production. In other words, we suggest that this is a consequence of the chemistry rather than the cause of the $Cl_2$ production.

Our observations suggest that photochemistry on the particle surfaces is the important driver of the high $Cl_2$. The strong correlation between $P(Cl_2)$ and the product of nitrate and aerosol surface area density suggests that photolysis of nitrate-laden particles may be involved in the chloride activation to produce $Cl_2$ at our site. (Note that the correlation was largely decreased ($R^2 = 0.39$) if the surface area density was excluded). The $Cl_2$ production via chloride activation also requires particulate chloride. Interestingly, the average chloride ($Cl^-$) concentrations were comparable in the oceanic air with low $Cl_2$ (0.56 µg m$^{-3}$ in $PM_{2.5}$ and 2.47 µg m$^{-3}$ in $PM_{10}$) and the continental air mass (0.50 µg m$^{-3}$ in $PM_{2.5}$, 2.38 µg m$^{-3}$ in $PM_{10}$). The average Cl/Na mass ratio in the oceanic air was 1.48 in $PM_{2.5}$ and 1.63 in $PM_{10}$ compared to 1.10 in $PM_{2.5}$ and 1.33 in $PM_{10}$ in continental air, indicating that Cl was more depleted in polluted air than in the clean air, in comparison to their average ratio of 1.8 in seawater[30]. These results suggest that $Cl^-$ was not the limiting factor, and the $Cl_2$ production was mainly controlled by nitrate availability and other factors.

**Laboratory investigations of $Cl_2$ production.** To explore the photochemistry leading to $Cl_2$ production, a series of experiments were undertaken by illuminating nitrate and chloride-containing solutions and ambient aerosols in the presence of gaseous $O_3$ with a high-pressure xenon lamp (Supplementary Fig. 6). The experimental setup and detailed information (designs, lamp, and chemicals) are given in Methods section "Lab Experiments" (and Supplementary Fig. 7). The average relative humidity (RH) during the field measurements was 81%, which was above the deliquescence point of sodium chloride (75%), and thus a very large fraction of sea-salt aerosols should have been wet during our field study. We, therefore, investigated $Cl_2$ production over or in solutions.

No $Cl_2$ was observed in the blank experiments, which were run with an empty chamber or with a quartz petri dish containing deionized water or chloride placed in the chamber, in the dark, or when illuminated by the xenon lamp. We also did not detect $Cl_2$ when zero air containing various $O_3$ mixing ratios (150, 250, and 500 ppbv) flowed over the illuminated solution of 1 M sodium chloride. The result shows that $O_3$ photolysis alone does not produce any detectable amount of $Cl_2$ in our experiment, as observed previously[27]. We note that the rate constant for $O_3$ to generate $O(^1D)$ ($1.31 \times 10^{-5}$ s$^{-1}$) in our experiment was two orders of magnitude lower than that in the previous study[27], who used a more intense UV light source.

Interestingly, we observed significant $Cl_2$ production when acidic solutions (pH < 3.3) containing both chloride and nitrate were illuminated. Irradiation of the solution, with an initial pH of 1.9, led to a continuous increase of gaseous $Cl_2$, and up to 3.5 ppbv was observed after 500 min of illumination (Fig. 2A). Previous laboratory studies of halogen production under similar but not identical conditions (i.e., light source and reaction medium) indicated that reactive bromine gases ($Br_2$ and BrCl) were produced over acid-doped nitrate-halide solution (liquid or frozen) under UV light (~310 nm)[31–33], but $Cl_2$ was not observed, unlike our experiment. Note that in our study, $Br_2$ and BrCl were also produced together with $Cl_2$. We also investigated the influence of ozone on $Cl_2$ production. There was no relative

increase in the $Cl_2$ signals when zero air containing differing $O_3$ mixing ratios (150, 250, and 500 ppbv) flowed over the illuminated chloride-nitrate solutions with a pH of 1.9 to 2.9, compared to the no $O_3$ cases (Supplementary Fig. 8A for pH = 1.9). And the $Cl_2$ level also did not increase when the added $O_3$ increased from 150 ppbv to 500 ppbv with a pH of 3.3 to 6.8 (Supplementary Fig. 8B for pH = 3.9). When we placed an AM1.5 optical filter in front of the xenon lamp, which only allows the light with the wavelength > 360 nm to pass through, there was a sharp decrease in the $Cl_2$ (and HONO) signals (shown at t = 540 min), whereas using a 300–800 nm optical filter (allowing the 300–800 nm light to pass through) only slightly decreased the $Cl_2$ (and HONO) production (shown at t = 520 min). This result reveals large $Cl_2$ production occurring at the wavelength of < 360 nm despite its concurrent significant photolytic loss.

Based on the above results, we propose that the hydroxyl radical (OH) from the nitrate photolysis and subsequent oxidation of chloride in solution was primarily responsible for the observed high rate of $Cl_2$ production (R12–R14, R4, R5).

$$NO_3^- + h\upsilon\,(<350\,nm) \rightarrow O^- + NO_{2(aq)}\,(R1,\,yield \sim 0.01)$$
(R16)

$$\rightarrow NO_2^- + O(^3P)_{(aq)}\,(R2,\,yield \sim 0.001)$$ (R17)

$$O^- + H^+ \rightleftharpoons OH_{(aq)}$$ (R18)

It has been known that nitrate absorbs light in the actinic range of 290–350 nm and dissociates via two pathways (R16 and R17) with a quantum yield of 0.017 and 0.001, respectively[34–37]. $O^-$ (produced from R16) reacts with water to form the hydroxyl radical (OH) (R18). This process can be accelerated by the acidity of the solution[34]. The produced OH can further oxidize $Cl^-$ to produce $Cl_2$ in the liquid phase (R4-R9), according to previously known aqueous chemistry[28,34,38], and a portion of the $Cl_2$ is released to the gas phase. Our observation of HONO and $NO_2$ production (see Fig. 2A, B) supports that R16 and R17 were taking place in our experiment and is consistent with the previous studies showing the production of HONO and $NO_2$ from illuminated nitrate solutions[39]. To confirm the role of aqueous OH radical in the $Cl_2$ production, we added 10 µl 0.1 M Tert-Butyl Alcohol[40] (TBA, a scavenger of OH radical scavenger with a rate constant of $(3.8–7.6) \times 10^8$ M$^{-1}$ s$^{-1}$) in the illuminated solution (Fig. 2B). There was a sharp decrease of the $Cl_2$ signal lasting for 20 min before returning to the previous level. To make sure that this change was not caused by operation (i.e., opening the chamber), we added 10 µl Deionized (DI) water into the chamber, and the $Cl_2$ signal bounced back in a few seconds. This result confirmed that the aqueous OH radical played a significant role in the $Cl_2$ production in the chamber.

We found that the $Cl_2$ production was strongly dependent on the acidity of the solution (Fig. 2C and Supplementary Fig. 9). The production rate sharply dropped to near zero when the pH increased from 2.9 to 3.3. It is expected that increasing pH would decrease the OH radical production via R18 and through the $HOCl^-$ adduct (R3–R9), and hence decrease the rate of $Cl_2$ production via R4–R9[28]. In addition, when the pH increases ($H^+$ decreasing), nitrite ions ($NO_2^-$) would produce less HONO in the aqueous phase, which in turn produces less aqueous OH radical (via R19) and then $Cl_2$. Interestingly, there seems a critical pH (~3.3) above which little $Cl_2$ is produced. This can be explained by suppression of OH by $NO_2^-$ above this pH value of 3.3. The dissociation constant (pKa) of HONO at 298 K is 3.3[41,42], i.e., above pH = 3.3, $NO_2^-$ is the predominant species in solution. We found that $NO_2^-$ can efficiently suppress OH concentration. When we added a very small amount of $NO_2^-$

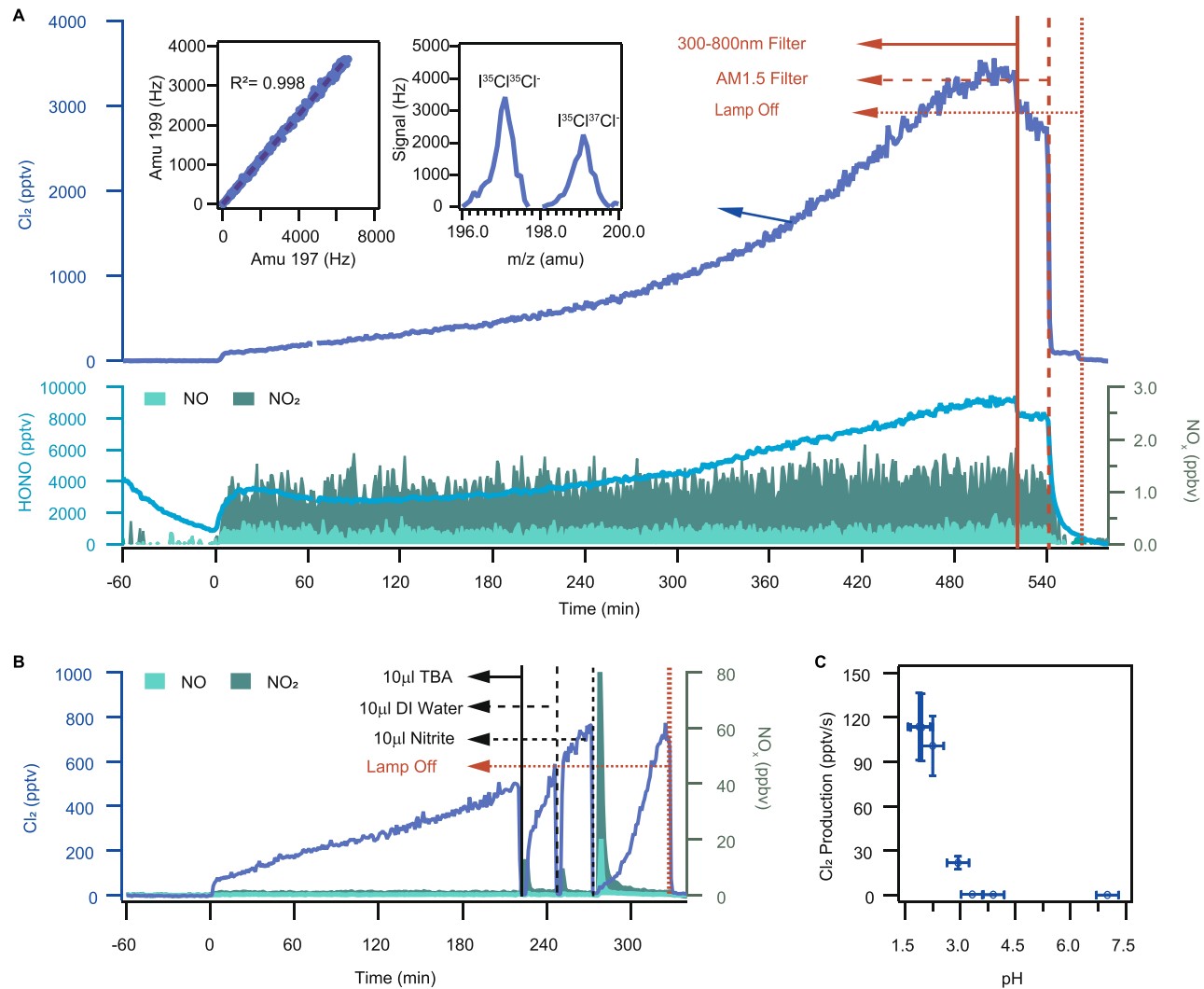

**Fig. 2 Experimental results on solutions in the dynamic chamber. A** Time series of 1-min average mixing ratios of $Cl_2$, HONO, and $NO_x$. The liquid solution samples (pH = 1.9) were illuminated at $t = 0$. The solid red line shows the time at which 300–800 nm filter was used, the red dashed line indicates the time at which AM1.5 filter was used, and the red point line indicates the time at which the xenon lamp was turned off. The left inset: scatter plot of the raw CIMS signal of $Cl_2$ at mass 199 atomic mass unit (amu) ($I^{35}Cl^{37}Cl^-$; $I^{37}Cl^{35}Cl^-$)) versus 197 amu ($I^{35}Cl^{35}Cl^-$) with 1-min average from $t = -60$ to $t = 580$ min. The right inset: the scanned mass spectra from 196 amu to 200 amu at $t = 387$ min. The continuous increase of $Cl_2$ may be due to the higher concentration of ions and acidity in the solution due to the evaporation of water from the solution. **B** Time series of 1-min average $Cl_2$, NO, and $NO_2$. The liquid solution samples (pH = 2.0) were illuminated at $t = 0$. The solid black line shows the time at which 10 µl OH scavenger, TBA, was added, the black dashed line indicates the time at which 10 µl DI water was added, the black point line indicates the time at which 10 µl nitrite was added, and the red point line indicates the time at which the xenon lamp was turned off. **C** The production rate of $Cl_2$ as a function of initial solution pH (pH = 1.9; 2.0; 2.3; 2.9; 3.3; 3.9; 6.8) at the illumination time of 500 min. The error bars in the plot (**C**) represent the estimated uncertainty in $Cl_2$ and pH measurement. Experimental conditions: 75−83% RH, 298 K in air and one 4 ml liquid solution containing 1 M NaCl + 1 M $NaNO_3$.

(10 µl 0.01 M) in the illuminated solution, the concentration of $Cl_2$ decreased significantly (Fig. 2B), revealing that $NO_2^-$ is an OH scavenger. Figure 2B shows that it took twice as long for the $Cl_2$ signal to return to the previous level, compared to the case of TBA, suggesting that $NO_2^-$ is a more efficient OH scavenger than TBA. In summary, based on our experimental results, we hypothesize that the photolysis of nitrate has two different effects on $Cl_2$ production. One is to promote $Cl_2$ production by increasing OH (R4–R9), and the other is to inhibit $Cl_2$ formation via nitrite. Increasing solution pH allows more $NO_2^-$ to stay in the solution and reduces $Cl_2$ production.

$$NO_2^- + H^+ \rightleftharpoons HONO\,(aq) \rightleftharpoons HONO\,(g) \qquad (R19)$$

We also investigated the effect of surface area on $Cl_2$ production. In the laboratory experiments, more $Cl_2$ was observed when 4 mL

of the nitrate-NaCl solution was split into $4 \times 1$ mL samples (Supplementary Fig. 9). This may be explained by increased $Cl_2$ production and diffusion into the gas phase from the increased surface area. Previous kinetic modelling[28] and laboratory studies[43] indicated preferential occupation of nitrate ions at the interface, which can facilitate fast surface reactions.

To further investigate the daytime $Cl_2$ formation under ambient conditions, four aerosols samples collected (for 24-h duration each) at the same site on 11–13 October 2020 were irradiated in the dynamic chamber. As shown in Fig. 3A and Supplementary Table 2, $Cl_2$ mixing ratios of up to 600 pptv were observed after illuminating two of the aerosol particle-loaded filters (filter 01 and 02) containing high concentrations of $Cl^-$ and $NO_3^-$. Interestingly, the produced $Cl_2$ were below the detection limit in the other two filters (filter 03 and 04) loaded

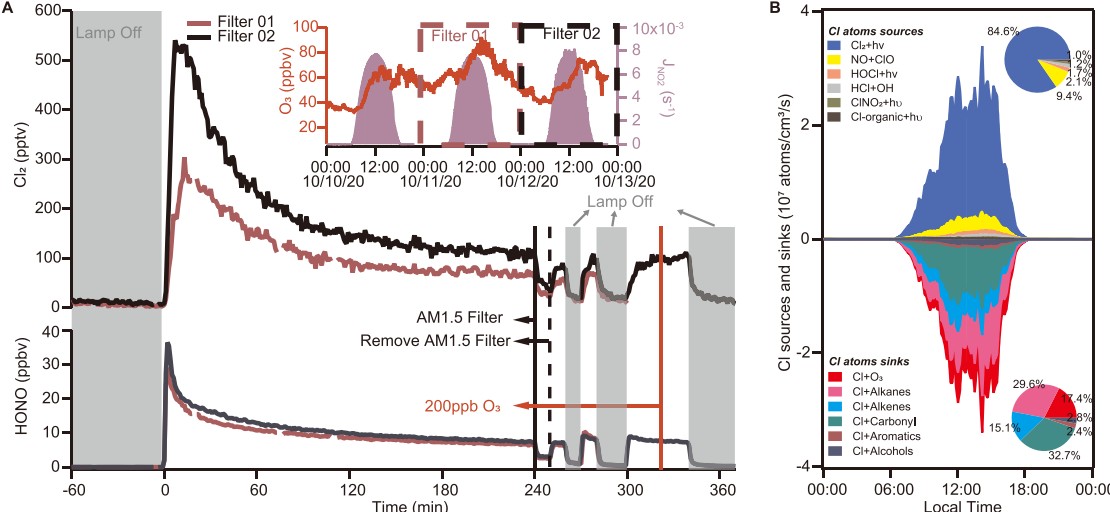

**Fig. 3 Experimental results on ambient aerosols and model results of Cl atoms budget. A** Experimental results on ambient aerosols in the dynamic chamber. Time series of 1-min average $Cl_2$ and HONO. Ambient samples (cropped size: 60 mm × 60 mm, Fig. S7C) were illuminated at $t = 0$. The grey area indicates the period, which the xenon lamp was turned off, the solid black line shows the time at which the AM1.5 filter was used, the black dashed line indicates the time at which the AM1.5 filter was removed, and the solid red line indicates the time at which 250 ppbv $O_3$ was added. The right inset: ambient observations of $O_3$ and $J_{NO2}$ during the ambient aerosol collection in October 2020. Experimental conditions: 75—83% RH and 298 K in air. **B** The model-calculated average diurnal profiles of sources and sinks of the Cl atom for period 4—14 September 2018. Upper right inset: the daytime average contribution from different sources to Cl atom. Bottom right inset: the average daytime contribution from different sinks to Cl atom.

with particles that contain low concentrations of $Cl^-$ and $NO_3^-$. In filter 01, the high level of $Cl_2$ was observed along with HONO, suggesting the potential role of particulate nitrate photolysis in their productions. Similar to the experiment performed on the $Cl^-$ and $NO_3^-$ solution, the $Cl_2$ levels decreased with the use of AM1.5 optical filter, indicating the wavelengths < 360 nm are very important for $Cl_2$ (and HONO) production, and no increase in $Cl_2$ was observed when we flowed the zero air containing 250 ppbv $O_3$ (Fig. 3A).

We next attempt to extrapolate the laboratory results to account for the observed atmospheric daytime $Cl_2$. The field observations and laboratory experiments suggest that the $Cl_2$ production likely occurs on the aerosol surface. The pH for most aerosol samples (>90%) in the 2018 field campaign was in the range of 1–3, with an average value of 1.5 (Supplementary Fig. 10), which was estimated from the E-AIM model (see Methods section "HYSPLIT and E-AIM models"). The laboratory-determined $Cl_2$ production rates on liquid solutions at pH of 1.9 (similar to the average value of the ambient aerosol), and 1 mol $L^{-1}$ nitrate was 114 pptv $s^{-1}$ (shown at $t = 520$ min in Fig. 2). The surface area density of the solution in the chamber air is $5.13 \times 10^5$ $\mu m^2$ $cm^{-3}$ (see Methods section "Lab Experiments"), which gives a $Cl_2$ production rate of 214 mol $m^{-2}$. For the continental air mass and during the period of 10:00–15:00, the average surface area density of ambient aerosols was 653 $\mu m^2$ $cm^{-3}$ after taking into account aerosol hygroscopicity when assuming the particles to be spherical, and the nitrate concentration in the aerosol liquid phase estimated by the E-AIM model was 3.9 mol $L^{-1}$. This gives an estimated $Cl_2$ production of 0.57 pptv $s^{-1}$, which could explain 68% of the observed average $Cl_2$ production rate (~0.84 pptv $s^{-1}$) in the ambient air. In addition to the airborne aerosol, the $Cl_2$ production may occur on the aerosols deposited on the ground, which could provide additional production that would help to reconcile the lab and field observed $Cl_2$ productions. We note that the above extrapolation is subject to uncertainty, including that from applying the $Cl_2$ production over the tested solution to the ambient aerosol, that from estimating aerosol pH with current aerosol thermodynamic models, and not accounting for

competing reactions for OH and Cl, such as by organics. It is also possible that other unidentified source(s) may contribute to part of the observed daytime $Cl_2$.

The above results indicate the importance of the coexistence of the three key factors in $Cl_2$ production, namely, nitrate, chloride, and acidity in the aerosol particles. Our proposed $Cl_2$ production mechanism could qualitatively explain the lack of daytime $Cl_2$ in previous studies that reported lower aerosol acidity or lower aerosol chloride content. During shipborne measurements off the coast of Los Angeles, elevated $Cl_2$ concentrations were observed mostly at night and in isolated industrial plumes[15]. Their E-AIM model calculated sub-micrometer aerosol pH was > 4. According to the pH-dependency of $Cl_2$ production in our experiments, the $Cl_2$ production is expected to be very slow under such conditions, which could explain the absence of daytime $Cl_2$ in their study. Very low levels of $ClNO_2$ (and lack of $Cl_2$) were reported in the oil-exploration impacted Uintah basin[44], and the author attributed the low $ClNO_2$ to lacking aerosol-phase chloride at this remote inland site, which would also explain the absence of $Cl_2$ in that study.

Our result has shown that pH below 3.3 is a necessary condition for nitrate-induced $Cl_2$ production. There could be a lower limit of pH for this production pathway. If the pH were too low, $Cl^-$ would be converted to HCl and $NO_2^-$ to HONO, the former would reduce $Cl_2$ production while the latter would promote $Cl_2$ formation. Accordingly, there could be a window of suitable pH conditions for the described way of $Cl_2$ formation to efficiently take place. The estimated average pH for $PM_{2.5}$ at our study site was 1.6 (range: 0.5–3.0), and a very large production of $Cl_2$ was observed under such acidic conditions. A recent review of global aerosol acidity[45] indicates that about 90% of fine-mode aerosol has a pH larger than 0.5, with 80% above 1.6. We suggest that the lower pH limit at which $Cl_2$ production would decrease may not be reached for typical tropospheric aerosols. That is, aerosol pH smaller than 3.0 should promote $Cl_2$ production under most tropospheric conditions.

**Impact on atmospheric chemistry and implications.** We assess the effects of the observed $Cl_2$ on VOC oxidation using a

photochemical box model with up-to-date VOC-Cl chemistry[6] (also see Methods section "Chemical box model") by constraining the model with the measured $Cl_2$ abundance and other relevant observations for 4–14 September 2018. The average values and the diurnal profiles of the input data indicate moderately polluted conditions for this period, with an average peak mixing ratio of $O_3$ of ~80 ppbv and $Cl_2$ of 300 pptv and $NO_x$ and VOCs levels characteristic of the polluted rural environment (Supplementary Table 1, and Supplementary Fig. 4). The model predicted that Cl atoms reached a maximum concentration of $~2.5 \times 10^5 \, cm^{-3}$ at noon (Supplementary Fig. 11A), with $Cl_2$ photolysis being the dominant source (~85%) (Fig. 3B). The peak Cl production rate at our site ($~4 \times 10^7 \, cm^{-3}s^{-1}$, Fig. 3B) is more than five to six times of that from the photolysis of BrCl and $Cl_2$ in winter[6] or from the photolysis of $Cl_2$ (predominantly) and $ClNO_2$[19] in summer in a rural area of northern China, and is two orders of magnitude larger than that from the photolysis of $Cl_2$ and $ClNO_2$ in late autumn and early winter at a ground site near the City of Manchester, UK[20].

The Cl atoms accounted for 59% of daily integrated oxidation of non-methane alkanes, 16% of aromatics, 13% of aldehydes, and 9% of dialkenes (Supplementary Fig. 12). The reactions of Cl atoms with VOCs produce $RO_2$ radicals, which are recycled to form $HO_2$ and OH radicals, thereby collectively increasing the average mixing ratios of OH, $HO_2$, $RO_2$ radicals by ~4%, ~17%, and ~27%, respectively (Supplementary Fig. 11). The enhanced $HO_2$ and $RO_2$ by Cl-VOC reaction increased the in-situ net total ozone ($O_x$, $O_3 + NO_2$) production rates by 17% (or 1.6 ppbv $h^{-1}$) and its daily integrated production by 16% (or 38 ppbv $day^{-1}$) (Supplementary Fig. 11), despite destroying ozone by Cl atoms at the same time (Fig. 3B). With a high-resolution time of flight mass spectrometer, we also observed elevated concentrations of organochlorides (e.g., 1-chloro-3-methyl-3-butanone, CMBO) with a similar diurnal profile to $Cl_2$, a possible indication of significant oxidation of VOCs by chlorine atoms (Supplementary Fig. 13). These results demonstrate the substantial impact of $Cl_2$ on daytime oxidation chemistry at our moderately polluted site.

In summary, a limited number of prior studies have indicated the presence and important role of daytime $Cl_2$ in the photochemistry of the lower troposphere in polluted regions. However, the exact source or production mechanism remained uncertain, which has hindered the reproduction of the daytime $Cl_2$ in current state-of-the-art global and regional chemistry transport models. In the present study, we observed very high $Cl_2$ concentrations at a polluted coastal site, implying that the $Cl_2$ could exist in more places and at higher concentrations than previously thought, in view of the limited $Cl_2$ observations to date. Combining laboratory and field measurements, we show that $Cl_2$ can be produced from photolysis of aerosol containing nitrate, chloride, and high acidity, and demonstrate that this mechanism can explain a great fraction of the observed daytime $Cl_2$ at our measurement site. Our result indicates the critical role of aerosol acidity (pH < 3.3) in promoting $Cl_2$ production. The same mechanism may occur in other parts of the world impacted by anthropogenic pollution. Despite a significant reduction in the emissions of acid precursors like sulfur dioxide ($SO_2$) and nitrogen oxides ($NO_x$), highly acidic aerosols are still present in some areas/seasons in Asia, North America, and Europe[45], and nitrate aerosols are also abundant in world's urban and industrial regions[46,47]. Previous studies have also indicated the ubiquity of aerosol chloride in continental as well as maritime environments[3,30,48,49]. We, therefore, anticipate that the $Cl_2$ production operates in some places or times where/when sufficiently high levels of acidity, nitrate, and chloride co-exist. Our recent study in northern China also shows that nitrate photolysis could activate chloride and bromide in coal-burning

aerosol, which exerted a large impact on winter oxidation chemistry[6]. We note that elevated nitrate and the other acidic aerosol (sulfate) have been observed during the Arctic haze events[50,51]. It would be of great interest to investigate whether the nitrate photolysis mechanism would contribute to the liberation of inert chlorine in the polar troposphere.

Our findings have indicated a previously unrecognized role of the reactive nitrogen cycle in both halogen and $HO_x$ chemistry and, at the same time, an interesting coupling between condensed phase oxidation resulting in the formation of its gaseous counterpart, which is expected to have important implications on atmospheric chemistry and production of secondary air pollutants. These findings suggest a direction for developing the $Cl_2$ scheme to predict its impact on oxidation chemistry for air quality models that currently do not include such chemistry. Moreover, our results suggest an additional benefit of widely adopted antipollution measures to control $SO_2$ and $NO_x$. That is, reducing $SO_2$ and $NO_x$ emissions not only alleviates their adverse impacts on health and welfare directly and through the formation of acid deposition and particles, but also decreases particle acidity and nitrate, both of which would slow the $Cl_2$ production and its promoting consequences to secondary pollution production, for example, surface ozone. We call for more investigations of the roles of halogen chemistry in the polluted troposphere and suggest some research that would place $Cl_2$ (and other halogens) production and atmospheric impact on a firmer footing. They include more atmospheric measurements of $Cl_2$ together with aerosol acidity (or its proxies), nitrate, and other parameters in diverse geographical areas, detailed laboratory measurements of the photolysis of nitrate ion in aerosol as a function of acidity, complete characterization of the aerosol particles to identify the origin of chloride in them, robust ways to determine rates of photolysis in and on aerosol particles, and parameterization of $Cl_2$ production to assess the broader impact of reactive chlorine chemistry in regional and global models.

## Methods

**Field measurements**. $Cl_2$, $ClNO_2$, HONO, NO, $NO_2$, $O_3$, volatile organic compounds (VOCs), oxygenated volatile organic compounds (OVOCs), aerosol compositions (including/e.g., $Na^+$, $NH_4^+$, $SO_4^{2-}$, $NO_3^-$, $Cl^-$), solar radiations, and other meteorological parameters were measured from 31 August to 09 October of 2018 at the Hong Kong Environmental Protection Department's Cape D'Aguilar Super Site, which is situated at the southeast corner of Hong Kong Island (Supplementary Fig. 1). We introduce in detail $Cl_2$ and other species measured by a chemical ionization mass spectrometer (CIMS). Information on other measurements is summarized in Supplementary Table 3.

Reactive chlorine species (including $Cl_2$, $ClNO_2$, and HOCl) and HONO were measured by a quadrupole CIMS (Q-CIMS). A detailed description of the CIMS and ion chemistry has been described in our previous studies[6,17]. Briefly, Iodide ($I^-$) was used as a reagent ion. $Cl_2$ was monitored at 197 amu ($I^{35}Cl^{35}Cl^-$) and 199 amu ($I^{35}Cl^{37}Cl^-$), $ClNO_2$ at 208 amu ($I^{35}ClNO_2^-$) and 210 amu ($I^{37}ClNO_2^-$), HOCl at 179 amu ($IHO^{35}Cl^-$) and 181 amu ($IHO^{37}Cl^-$), and HONO at 174 amu. In this study, we used the data of $Cl_2$, $ClNO_2$, and HONO from CIMS measurement. The HOCl signals suffered from spectral interference, as indicated by the weak correlation between the two isotopic masses, and thus were not used in further analysis.

The instruments were housed in a one-story building. The inlet is a 3.5-m long PFA-Teflon tubing (1/2 in. outer diameter) with 1.5 m of it situated above the roof. We adopted the previous inlet design as described in our previous study[6]. To further reduce the residence time (and thereby potential artifacts) in the inlet tubing, we used a blower with a flow rate of 500 SLPM flow to draw the sample. As a result, the residence time of the sample air in the inlet tubing was below 0.1 s. To reduce the particle deposited on the inlet tubing, the tubing was flushed with DI water and then dried by drawing ambient air for 20 min every three days.

The following on-site and post-measurement measures were undertaken to ensure accurate measurements of $Cl_2$. They included (1) Instrument background determination. During the study, the CIMS background signals were determined about every two days by scrubbing ambient air with alkaline glass wool and charcoal[6]. Many inorganic halogens are efficiently removed by this process. In the present study, the background for $Cl_2$ was small and relatively stable at around 5 pptv during the field campaign (Supplementary Fig. 14B). The 2-σ detection limit was 9 pptv for $Cl_2$ (at 197 amu). (2) Regular calibration with a $Cl_2$ standard. The

calibration of $Cl_2$ was conducted on-site every 2–4 days with a $Cl_2$ permeation tube. The detailed calibration methods have been described in our previous study[6]. The permeation rate of the $Cl_2$ standard was determined before and after the campaign and was stable at around 210 ng min$^{-1}$, with a variation of less than 5% during the field campaign. The sensitivity of $Cl_2$ was stable at 2.0 Hz pptv$^{-1}$ with a standard deviation of 0.2 Hz pptv$^{-1}$, as shown in Fig. S14D. The $Cl_2$ sensitivity was invariant at RH of 20–80% (Supplementary Fig. 14E). The measurement uncertainty for $Cl_2$, calculated from the variation of the sensitivity during the campaign and the uncertainty of permeation tube source, was about 11%. (3) Examination of the isotopic ratio of the detected compounds. During the field study, we confirmed that the detected signal for $Cl_2$ had no significant spectral interference. The two isotopic masses at 197 amu and 199 amu were well resolved, as shown in Supplementary Fig. 14A, and showed excellent correlation ($R^2 = 0.93$) with a slope of 0.63 (Supplementary Fig. 14C), which is similar to the theoretical value of 0.65. (4) Investigation of conversion on inlet surfaces. We conducted a series of tests to examine potential inlet interferences in the field and laboratory, which included: (i) potential inlet artifact arising from heterogeneous reactions of ambient $O_3$ and $N_2O_5$ were tested by adding $O_3$ and $N_2O_5$ into the ambient air in the intake to the measurement system. $Cl_2$ often coincided with high $O_3$ concentrations in the field, and previous lab[52] and field studies[17] indicate potential $Cl_2$ formation involving $N_2O_5$. During the field sampling period, when we turned off the bypass blower and injected concentrated $O_3$ and $N_2O_5$ into the ambient air sample (resulting in 250 ppb of $O_3$ and 5 ppbv of $N_2O_5$ after mixing with the ambient air), we did not observe any increased $Cl_2$ signals. This result indicates that the $O_3$ and $N_2O_5$ did not produce detectable artifacts in our $Cl_2$ measurement. (ii), potential inlet artifact tests of HOCl reactions in the laboratory. A previous study reported that 15% of the HOCl was lost in their NaCl-coated inlet, with 2% converting to $Cl_2$[11]. Our post-campaign tests confirmed that the observed $Cl_2$ did not suffer from significant interference from HOCl in the sampling inlet. Briefly, we tested two types of Teflon tubing: one used in the campaign and a new tubing with the same length. HOCl was synthesized using a phosphate-buffered solution (pH = 6.8) of NaOCl (11–14% chlorine, Alfa Aesar) and AgNO$_3$, analogous to the previous method[11,14]. A 20 sccm dry $N_2$ was flowed through the solution and then diluted into 6 SLPM humidified zero air. The concentration of HOCl was calculated from the $Cl_2$ formation by passing the HOCl standard through a NaCl-coated tubing. For the HOCl inlet conversion test, the synthesized HOCl mixed with 6 SLPM humidified zero air was first introduced to the CIMS without passing through the tubing. Then, the HOCl/air mixture passed through the tubing before entering the CIMS. The decrease in the HOCl signal and the increase in the $Cl_2$ signal induced by the tubing were monitored to determine the conversion of HOCl to $Cl_2$ in the tubing. Under the RH condition similar to the field campaign, we found that 31% and 7% of the HOCl were lost, and 18% and 2% were converted to $Cl_2$ in the used tubing and the new tubing, respectively. As the flow rate in the laboratory (6 SLPM) was much lower than the ambient sample flow rate (500 SLPM), we conclude that the conversion rate of HOCl to $Cl_2$ during the field measurement should be much lower than 18%.

**HYSPLIT and E-AIM models**. Three-day (72 h) backward trajectories were calculated for each hour using the Hybrid Single-Particle Lagrangian Integrated Trajectory (HYSPLIT) model (https://www.ready.noaa.gov/HYSPLIT.php). The HYSPLIT was driven by 3-hourly archive data from NCEP's GDAS with a spatial resolution of 1 degree by 1 degree. The endpoint of the trajectories was 300 m above ground level at Hok Tsui, which is in the middle of the marine boundary layer. Air masses were then classified based on the source regions (ocean or continent).

The $H^+$ concentrations ([$H^+$], in mol L$^{-1}$) in the aqueous phase of aerosols were calculated using the E-AIM model (E-AIM III) online (http://www.aim.env.uea.ac.uk/aim/model3/model3a.php)[17,53]. The inputs to the model are hourly measurements of ambient RH and molar concentrations (unit: mol m$^{-3}$) of $Cl^-$, $NO_3^-$, $SO_4^{2-}$, $Na^+$, and $NH_4^+$ in $PM_{2.5}$, which were measured by an ion chromatography (MARGA, Supplementary Table 3) and gas-phase ammonia. Aerosol pH was estimated as the negative logarithm of [$H^+$] without further consideration of the activity coefficient of ions in the aqueous phase.

**Chemical box model**. A zero-dimensional gas-phase chemical box model was used to calculate the budget for Cl atoms and to evaluate the observed $Cl_2$ on atmospheric oxidation. The detailed information on the mechanisms and their related kinetics data of gas-phase reactions adopted in the model is given in the previous study[6]. The measured values of $Cl_2$, $ClNO_2$, $N_2O_5$, HONO, $O_3$, NO, $NO_2$, $SO_2$, CO, and temperature were averaged or interpolated every minute and constrained into the model. The measured VOCs and OVOCs (except for $CH_4$ and HCHO) were interpolated every minute and constrained into the model. The mixing ratio of $CH_4$ was kept at a constant value of 2000 ppbv[54]. As the HCHO measurement data was not available in the 2018 field campaign, we used the HCHO measurement data obtained during September 2020 by off-line DNPH-Cartridge-HPLC (24h-average, 3.3 ppb) and adjusted for its diurnal variation according to a typical HCHO profile in a non-urban environment[55]. The input data for HCHO is shown in Supplementary Fig. 4.

The photolysis frequencies for $Cl_2$, HONO, $O_3$, and other species were calculated from the TUV model (http://cprm.acom.ucar.edu/Models/TUV/Interactive_TUV/) under clear sky condition and then scaled to $J_{NO2}$, which was derived from the measured solar radiation and relationship with $J_{NO2}$ for Guangzhou (~100 km north of the present site)[56]. The dry deposition process in the model was represented by a first-order loss reaction, using the same parameter described in the previous study[57]. The boundary layer height was set at 200 m at nighttime and 1500 m for daytime in the model. The wet deposition was ignored as no rainfall event occurred during the observation period. The model was run from 00:00 of 4 September to 00:00 of 14 September, and the simulation for the first 24 h was repeated three times to stabilize the intermediate species. A summary of the input parameters in the model is shown in Table. S1, and the diurnal patterns of select input data are shown in Supplementary Fig. 4.

**Estimation of $Cl_2$ production from heterogeneous reactions of HOCl**. $Cl_2$ can be produced from heterogeneous reactions of gaseous HOCl on a chloride-containing solution, with an uptake coefficient of HOCl up to 0.0002[22,58]. We used the following Eq. (1) to estimate the $Cl_2$ production rate from the HOCl heterogeneous reaction.

$$\text{The } Cl_2 \text{ production rate} = \frac{d[\text{HOCl}]}{dt} = \frac{1}{4} c_{\text{HOCl}} \gamma S_a [\text{HOCl}] \tag{1}$$

Where $c$ is the mean molecular speed of HOCl, $\gamma$ is the heterogeneous uptake coefficient of HOCl, [HOCl] is the model simulated concentration, and $S_a$ is the aerosol surface area density.

## Lab Experiments

*The laboratory design*. A dynamic reaction chamber was used to measure the productions of $Cl_2$ by illuminating nitrate-NaCl solution and aerosol collected on filters. The overall experimental setup is shown in Supplementary Fig. 7 and is described here. The chamber is made of TFE Teflon (1.875 L, 25 cm-length × 15 cm-width × 4 cm-height) with a TFE Teflon-film window on the top. A quartz petri dish (inner diameter: 35 mm, internal height: 7 mm) held 4-ml liquid solution or filter samples. The surface area density of the chamber was determined as the physical surface area of the solution in the petri dish divided by the chamber's volume and was $5.1 \times 10^5$ μm$^2$ cm$^{-3}$. Zero air (2.9 SLPM) with adjustable humidity (75–83%) flowed through the chamber. The experiments were conducted at room temperature (296 K). A flow of $O_3$ was diluted by zero air and then added into the chamber with the resulting $O_3$ mixing ratio in the chamber ranged from 0 to 500 ppb. The residence time of the zero air/$O_3$ was 0.625 min in the chamber. The outflow of zero air carrying the reaction products was monitored in real-time by the same iodide-CIMS instrument used in the field for $Cl_2$ (amu 197, 199) and HONO (amu 174) detection and by a chemiluminescent/photolytic converter for NO and $NO_2$.

To mimic the spectrum of the solar radiation, a high-pressure xenon lamp was used as the light source, and its spectral irradiance is shown in Supplementary Fig. 6. It covers from 320 nm to 1100 nm and peaks at 450 nm. Compared to the solar irradiance at a solar zenith angle of 48.2° (i.e., an air mass factor of 1.5 and standard ozone column abundance), the xenon lamp has a smaller flux in the range of 300 nm–326 nm but a larger flux in the range of 326 nm–420 nm. The photolysis rate constant for $O_3$ to generate $O^1D$ ($1.31 \times 10^{-5}$ s$^{-1}$) was similar to the daytime averaged rate constant of $1.78 \times 10^{-5}$ s$^{-1}$ (calculated from the TUV model under clear sky conditions) in the ambient air at our site (see Methods section "The determination of the photolysis rate and production rate of $Cl_2$"), and the photolysis rate constant of $Cl_2$ ($J_{Cl_2}$) in our chamber ($5.80 \times 10^{-3}$ s$^{-1}$, see below) was about four times larger than daytime averaged rate constant of $1.20 \times 10^{-3}$ s$^{-1}$ (calculated from the TUV model under clear sky conditions) in ambient air at our site. To investigate the role of photon energies, two optical filters were used (one is a 300–800 nm filter, which let the light with a wavelength of 300 - 800 nm to pass through, and the other, AM1.5 filter, which allows the light with the wavelength > 360 nm to go through).

To investigate the potential production of $Cl_2$ in chloride and nitrate-containing solution, sodium chloride (NaCl, ACS, > 99.8%) and sodium nitrate (NaNO$_3$, Sigma-Aldrich, >99.0%) were used as the source of particulate chloride and nitrate, respectively. Both NaCl and NaNO$_3$ were prepared as 1 M L$^{-1}$, which was similar to the average concentration of aqueous phase chloride and nitrate in ambient aerosols in the field study, which was estimated from the E-AIM (see above). The pH was adjusted by adding sulfuric acid (H$_2$SO$_4$, Sigma-Aldrich, 95–97%) and measured with a digital pH meter (HANNA instrument, HI253). In the experiments on the ambient filters, the aerosols of $PM_{2.5}$ collected on quartz fiber filters with a high-volume sampler (Flow: about 890 L min$^{-1}$, sampling period: 23.5 h, size: A4 page) were placed in the chamber.

*The CIMS measurements in the laboratory*. As was done in the field study, we conducted instrumental background checks, isotope analysis, and daily $Cl_2$ calibration in the laboratory experiments. The background for $Cl_2$ and HONO was stable. The sensitivity of $Cl_2$ was stable at around 1.9 Hz pptv$^{-1}$ with a standard deviation of 0.1 Hz pptv$^{-1}$. HONO was calibrated at the end of the lab experiment. The sensitivities for HONO during the laboratory studies were determined according to its sensitivity ratio relative to that for $Cl_2$. The sensitivity HONO was 3.0 Hz pptv$^{-1}$. The measurement uncertainty for $Cl_2$, calculated from the propagation of relative standard deviation for 1-min average data and the variation of the sensitivity within 1 day based on the calibration from permeation tube source, was about 5%. And measurement uncertainty for HONO, calculated from the

propagation of both relative standard deviation for 1-min average data and the variation of the sensitivity, was about 15%.

*The determination of the photolysis rate and production rate of Cl₂.* The photolysis rate of Cl₂ ($J_{Cl_2}$) in the chamber was calculated using the following Eq. (2)

$$J = \int q(\lambda)\sigma(\lambda)I(\lambda)d\lambda \qquad (2)$$

Where $q(\lambda)$ is the quantum yield at wavelength $\lambda$ (nm), $\sigma(\lambda)$ is the cross-section of Cl₂ at wavelength $\lambda$, which is adopted from the recommended value by IUPAC (http://iupac.pole-ether.fr/index.html). $I(\lambda)$ is the flux of xenon lamp at wavelength $\lambda$ and was calculated by converting the irradiation energy spectra of the lamp (Supplementary Fig. 6) to photon flux based on Planck's equation. The same method was used to calculate the photolysis rate constant for O₃ to generate O¹D. The $q(\lambda)$ and $\sigma(\lambda)$ was adopted from the recommended value from IUPAC under 298 K (http://iupac.pole-ether.fr/index.html).

All laboratory experiments were carried out under the same light intensity with the same distance from the chamber (20 cm). As shown in Supplementary Fig. 7, almost the entire bottom area is illuminated by light. Under this configuration, the $J_{Cl_2}$ was estimated to be $5.80 \times 10^{-3}$ s⁻¹ without the optical filter, which was around four times larger than the daytime averaged photolysis rate constant of $1.20 \times 10^{-3}$ s⁻¹ in the ambient air. In calculation of $J_{Cl_2}$ in the chamber, we did not consider light reflection at the Teflon window and in the chamber inner surface as well as light loss during transmission. The calculated $J_{Cl_2}$ was verified by another method, as shown in the end of this section.

The production rate of Cl₂ ($P_{Cl_2}$) in the dynamic chamber was determined based on the mass balance. The $P_{Cl_2}$ is equal to the sum of the photolysis loss rate of Cl₂ and the advected loss of Cl₂ in the dynamic chamber using the following Eq. (3):

$$P_{Cl_2} = \text{photolysis rate of Cl}_2 + \text{advected loss of Cl}_2 \qquad (3)$$

Thus, (Cl₂) (pptv s⁻¹) = [Cl₂] × $J_{Cl_2}$ + [Cl₂]× Q/V = [Cl₂]× ($5.8 \times 10^{-3}$ s⁻¹ + $2.7 \times 10^{-2}$ s⁻¹) under the experimental condition.

Where [Cl₂] is the measured Cl₂ mixing ratio (pptv), $J_{Cl_2}$ is the calculated photolysis rate (s⁻¹), $Q$ is the flow rate of the zero air thought the chamber (3 SLPM), and $V$ is the volume of the chamber (1.875 L). Equation (3) assumes negligible Cl₂ production from recombination of Cl atoms produced from Cl₂ photolysis in the chamber, and this assumption is verified by the following experiments: we compared the Cl₂ signals by the CIMS when 100 sccm a Cl₂ standard was diluted by 2.9 SLPM zero air and further mixed with 100 sccm zero air or 100 sccm ozone-containing zero air (yielding 500 ppbv ozone in the chamber air). These experiments were conducted without the aerosol or liquid film in chamber. There was no detectable change in the Cl₂ signals in the two tests (i.e., with or without ozone). This result confirms little Cl₂ production from Cl back reaction, as the Cl₂ signal with ozone added would have scavenged of Cl atom. Under the condition 3 SLPM flow (the condition of our experiments), advection was the predominant loss (accounting for 82%) of Cl₂ produced in the chamber.

To verify the calculated $J_{Cl_2}$, we compared the Cl₂ signals by the CIMS when the 100 sccm Cl₂ standard diluted by 2.9 SLPM zero air (yielding 3.25 ppbv Cl₂ in the chamber air) and then flowed through the chamber (without the condensed phase sample) with the lamp turned off and then on. The experimental results showed that there was a 16.5% drop in the Cl₂ signals with the lamp on. Using the above equation for $P_{Cl_2}$ Eq. (3), the $J_{Cl_2}$ was determined at $5.3 \times 10^{-3}$ s⁻¹, which is very close to the calculated value ($5.80 \times 10^{-3}$ s⁻¹) based on the lamp irradiance spectrum. Further, the calculated extent of Cl atom recombination is roughly consistent with this assertion of small loss of Cl atoms due to recombination. Based on these experiments and calculation, we suggest that the calculated $J_{Cl_2}$ is reliable.

## Data availability
The data that support the findings of this study are available in figshare with identifier (https://doi.org/10.6084/m9.figshare.17099252).

## Code availability
The HYSPLIT model can be acquired from the NOAA Air Resources Laboratory website (https://www.ready.noaa.gov/HYSPLIT_linux.php). The source code of the chemical box model is available from https://doi.org/10.6084/m9.figshare.17099252.

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

## Acknowledgements

We thank the Hong Kong Environmental Protection Department for providing space at its supersite for our field measurements and for the provision of some VOCs and aerosol composition data. We thank Chung Hon Larry Suen for the help in the field measurements. We thank Bobo Wang, W.S. Lam, Yanlin Zhang, and Prof. Wei Chu for their help in the laboratory experiments. This research is supported by the Hong Kong Research Grants Council (T24-504/17-N and A-PolyU502/16 to T.W.), the Agence Nationale de la Recherche (ANR-16-CE01-0013 to C.G.), the Swedish Research Council (2013-6917 to M.H. and C.M.S), and the European Research Council Executive Agency under the European Union s Horizon 2020 Research and Innovation programme (Project 'ERC-2016-COG726349 CLIMAHAL' to Q.L. and A.S-L.).

## Author contributions

T.W. and X.P. designed the $Cl_2$ research. T.W. and C.G. coordinated the field observations. X.P. designed the laboratory experiments with significant contribution from W.W., A.R.R., C.G., T.W., and Y.M. X.P., W.W., and M.X. performed halogen measurements. W.W., Y.W., and X.W. performed HONO measurements. W.W., C.N.P., and M.X. performed solar radiation field measurements. C.Y. performed aerosol size distribution measurement. M.C., A.M., Y.M., M.S., C.L., K.L., and A.Y. provided VOCs measurement data. M.C., A.M., Y.M., S.L., performed OVOCs measurement. C.M.S., and M.H. performed the organochlorides measurement with HR-Tof-CIMS. X.L., and H.G. collected ambient filters and performed ionic composition analysis. X.P. performed the laboratory work with the help from W.W., M.X., C.Y., C.N.P., Q. L., and Y.W. W.W. and X.P. conducted the chemical box model simulation. M.X. ran the E-AIM model for the calculation of the molar concentrations of inorganic ions. J.D. conducted the back-trajectory simulation. X.P., T.W., W.W., A.R.R., and C.G. analyzed the field, laboratory, and modeling results with significant contribution from Q.L., A.M., Y.M., M.H., A.S-L., H.H., and J.C. T.W. and X.P. wrote the paper with significant input from A.R.R., C.G., and A.S-L. All authors reviewed and commented on the final paper.

## Competing interests

The authors declare no competing interests.
