## [Peer Review File · Nature Communications]

Photodissociation of particulate nitrate as a source of daytime tropospheric Cl₂REVIEWER COMMENTS

Reviewer #1 (Remarks to the Author):

This paper presents some very unusual and intriguing observations of molecular chlorine at a site near Hong Kong. The results, if accurate, have some important implications for the atmospheric chemistry happening in near coastal semi-polluted sites. In principle, this work is important and novel enough to be published in Nature Communications. However, there are a number of questions and points of clarification that should be dealt with before the paper can be accepted for publication. I have the following general and specific comments.

General Comments:

The first thing that comes to mind when I read this paper is "why hasn't this been seen before?". Many of the older Cl₂ measurements were one-offs, with not a lot of other supporting information, and some like Spicer et al., (1998), simply don't pass the sniff-test when you look at the temporal profile of their Cl₂ signal. Since then, the community has fielded more extensive measurements of Cl species, perhaps the first big project was CalNex 2010, which has 3 different sets of measurements: ground aircraft and shipboard. That environment, the LA Basin, has substantial impacts from sea salt as well as NO_x pollution, yet high daytime Cl₂ was only occasionally observed, perhaps due to the chemistry discussed here, but many instances of high Cl₂ seemed to mostly be plumes from coastal industrial sources (see Riedel et al., 2012). The Uintah Basin Winter Ozone Studies (see Edwards et al., 2014) also had extensive Cl and NO_x measurement, but did not see this effect. I would like to see more discussion of why the authors think previous studies have not seen this effect, at least to this extreme extent. Is it confined to a narrow pH range? Were there not enough Cl⁻ sources?

What does the chemistry and associated box-modeling imply for levels of gas-phase HCl? I would imagine they have to be quite high in order to maintain a continuous source of soluble chloride to the particles. Was HCl measured, and what does the model require to make the chemistry work?

The reaction scheme for N and Cl species seems incomplete as presented, since we know that Cl₂ reaction on chloride-containing substrates can be used as a facile source of ClNO₂ (Thaler et al., 2011). This reaction is undoubtedly pH dependent, as it appears to go in the reverse direction below pH 2 (Roberts et al., 2008). Low pH conditions will of course liberate NO₂⁻ as HONO, and Cl⁻ as HCl, so could it be that there is a lower limit on the pH conditions at which this chemistry is happening? I would like to see more discussion of this.

There are several paragraphs where the authors stray off into making concluding remarks that seem out of place, and belong in the concluding section which is named "Impact on Atmospheric Chemistry and Implications". I will point those out below.

Specific Comments:

Line 57, 'VOCs' should be 'VOC'

Line 107-110. The chemistry as presented in R3-R5 is at least second-order in Cl⁻, (maybe 3rd-order?). Was that factored into the extrapolation?

Line 156-161. It seems to me that one key quantity here is the equivalent Cl⁻ molarity in the particles, assuming they are droplets. What were those numbers?

Fig. 1. I can't see the labels on the bottom of panels C&D.

Line 203 and Figure S8. This was really confusing as stated. Figure S8A shows increasing Cl₂, so when the authors said "no increase in Cl₂ signals", it took me a while to figure out they mean no relative increase with O₃ added compared to no O₃. The language needs to be clarified here. Also, the text here says pH 2.9 and the Figure says 3.9, so which is it?

Line 242-244. Did you see an increase in ClNO₂ anytime when adding NO₂?

Lines 264-276. Much of this material belongs in the concluding section.

Line 295-308. This paragraph belongs in the concluding section.

Line 350. What was the ionization chemistry of the HR-ToF-MS? I would assume proton-transfer-reaction (PTR), but that needs to be specified here.

Line 441. Did you observe any differences in signals or backgrounds after washing the inlet?

Line 462. I can't tell what is meant by "scrubbing the ambient air"?

Line 500. Don't you also need gas phase HNO₃ and HCl for the model.

Line 543. The number has too many significant figures.

References;

Edwards, P.M., et al., High winter ozone pollution from carbonyl photolysis in an oil and gas basin, *Nature*, 514, 351-354, doi:10.1038/nature13767, 2014.

Riedel, T.P., et al., Nitryl chloride and molecular chlorine in the coastal marine boundary layer, *Environ. Sci. Technol.*, 46, 10463-10470, 2012.

Roberts, J. M., H. D. Osthoff, S. S. Brown, and A. R. Ravishankara (2008), N₂O₅ oxidizes chloride to Cl₂ in acidic atmospheric aerosol, *Science*, 321, 1059., doi:10.1126/science.1158777.

Spicer, C.W., Unexpectedly high concentrations of molecular chlorine in coastal air, *Nature*, 394, 353-356, 1998.

Thaler, R. D., L. H. Mielke, and H. D. Osthoff, Quantification of nitryl chloride at part per trillion mixing ratios by thermal dissociation cavity ring-down spectroscopy, *Anal. Chem.*, 83, 2761-2766, doi:10.1021/ac200055z., 2011.

Reviewer #2 (Remarks to the Author):

This manuscript reports a combination of field measurements, laboratory experiments and modeling that document the photochemical production of Cl₂ in air during the day at mid-latitudes and the impacts on VOC oxidation etc. The conclusion is that nitrate ion photochemistry, which is well known to generate OH under acidic conditions, is responsible via OH production followed by its oxidation of chloride ions. The manuscript is well-written and it is comprehensive in its approach.

The major pieces are well-known: (1) nitrate photochemistry generates OH; (2) OH oxidizes halide ions such as chloride to form Cl₂; (3) Cl₂ photolyzes to generate Cl atoms that oxidize VOC, leading to chemistry similar to that initiated by OH radicals in the gas phase; (4) if the production is sufficiently fast compared to the rate of Cl₂ photolysis, measurable levels of Cl₂ build up during the day. With that background, the results reported are not surprising, but it is a nice combination of field, lab and modeling that quantifies these processes for this particular location and rules out other contributing processes as being important. The appropriateness of this paper for *Nature* rather than a specialty journal could be strengthened if the authors could better articulate the

unique and transformative aspects of the work.

Minor comments:

1. There is a long history regarding halogen chemistry in the troposphere. While the authors cannot cite every paper, perhaps the first specific measurement of tropospheric Cl₂ (Spicer et al, 1998) and some of the recent papers from the Pratt, Thornton, Bertram etc groups might be useful to readers.
2. There is a lot of literature on nitrate ion photochemistry with sea salt, or mixtures of chloride and bromide that mimic sea salt, e.g. Anastasio 2007, Abbatt 2010, Finlayson-Pitts, 2012. It appears that Br⁻ is oxidized first but when bromide is depleted, Cl₂ is observed, which is similar to what is reported here. Citing some of these studies might be helpful.
3. It was not clear to this reviewer why a stronger correlation occurs when particle surface area is included in the analysis. Their model results indicate that interface reactions R3-R5 cannot be important. Did they look at a correlation such as solar radiation x Cl⁻ x NO₃⁻, i.e. the major active species without the surface area? If it is nitrate photochemistry at the interface, is there likely to be enough nitrate at the interface? Could preferential occupation of ions at the interface (e.g. Richards-Henderson 2013) play a role?
4. Lines 191-199: the statement that "We did not detect Cl₂ when the zero air containing various O₃ mixing ratios... pH from 3.3 to 6.8. One example is shown in Fig. S8". But in Fig S8B, Cl₂ is observed at pH 3.9 as O₃ is increased. Clarification?
5. In the lab experiments, more Cl₂ is observed when the 4 mL of solution is split into 4 x 1 mL samples, and it is argued that this is due to an increase in surface area. However, could it be due to increased diffusion into the gas phase from the increased surface area?
6. Lines 295-308: Regarding the generality of the chemistry, the authors may want to include the possibility of chloride in road salt in winter climates as contributors (Ariya et al, 2020; papers by Pratt, Osthoff groups).
7. Figure S13: Some references to the identification of CMBO as a chlorine atom tracer (e.g. Nordmeyer et al, 1997; Tanaka et al, 2003) might be useful here.

Reviewer #3 (Remarks to the Author):

The authors measured high levels of Cl₂ in a coastal area in Hong Kong, the formation of which cannot be explained by known/recognized processes of Cl₂ formation from particulate chloride. Further analysis of the ambient data and laboratory experiments is consistent with Cl₂ activation via nitrate photolysis under acidic conditions. This topic is important and relevant to the field of atmospheric chemistry and air pollution, the work is original, methodologies used are sound, and the paper is generally written well. My main concern and only major comment is that I find the main results and implications a bit overstated. More specifically, the authors convinced me that nitrate photolysis under acidic conditions can activate Cl₂. However, I don't necessarily agree that they have "unraveled the daytime source of molecular chlorine in the extra-polar atmosphere" (quoting their title). The source of Cl₂ in the atmosphere is subject of ongoing studies across the world, and there are likely other processes of heterogeneous Cl₂ formation (from particulate Cl₂) and probably also several sources of Cl₂ emissions that are not accounted for in the models. This would also be consistent with the authors' observation that nitrate photolysis explains 13-68% (i.e. not all) of the observed Cl₂ production. I would thus suggest and request that the authors acknowledge the (in my opinion, high) likelihood of other processes and sources contributing to the high observed Cl₂ concentrations. They may have found an important source/process, but not unraveled the source. I suggest these changes be made before publication of the manuscript.

I have included a few more minor comments below.

General note: I likely did not catch all minor editorial errors/issues. Please re-review the manuscript for these possible errors.

Line 47: remove "a"

Lines 56-57: my understanding is that methane is not considered a VOC (e.g. the US EPA explicitly excludes it from its definition of VOCs)

Line 244: not clear what is meant by "lit solution"

Line 264: suggest removing "the" in front of "zero air"

Line 308: please review that line – there seem to be some errors in the prepositions used that make the sentence difficult to understand.

Response to Reviewer Comments #1

Reviewer #1 (Remarks to the Author):

This paper presents some very unusual and intriguing observations of molecular chlorine at a site near Hong Kong. The results, if accurate, have some important implications for the atmospheric chemistry happening in near coastal semi-polluted sites. In principle, this work is important and novel enough to be published in Nature Communications. However, there are a number of questions and points of clarification that should be dealt with before the paper can be accepted for publication. I have the following general and specific comments.

General Comments:

1. The first thing that comes to mind when I read this paper is “why hasn’t this been seen before?”. Many of the older Cl₂ measurements were one-offs, with not a lot of other supporting information, and some like Spicer et al., (1998), simply don’t pass the sniff-test when you look at the temporal profile of their Cl₂ signal. Since then, the community has fielded more extensive measurements of Cl species, perhaps the first big project was CalNex 2010, which has 3 different sets of measurements: ground aircraft and shipboard. That environment, the LA Basin, has substantial impacts from sea salt as well as NO_x pollution, yet high daytime Cl₂ was only occasionally observed, perhaps due to the chemistry discussed here, but many instances of high Cl₂ seemed to mostly be plumes from coastal industrial sources (see Riedel et al., 2012). The Uintah Basin Winter Ozone Studies (see Edwards et al., 2014) also had extensive Cl and NO_x measurement but did not see this effect. I would like to see more discussion of why the authors think previous studies have not seen this effect, at least to this extreme extent. Is it confined to a narrow pH range? Were there not enough Cl⁻ sources?

Response: Thanks for raising this important question. Yes, high daytime Cl₂ was not observed in some places such as the highly-urbanized LA basin indicated by the referee (Riedel et al., 2012, also cited in our manuscript). Based on the evidence in our present study, Cl₂ production is determined by at least three chemical factors: particulate-phase chloride abundance, nitrogen oxide availability and hence nitrate concentration in the aerosols, and high acidity. In particular, acidity seems to play a critical role.

Our Cl₂ production mechanism can qualitatively explain the lack of daytime Cl₂ in previous studies that reported aerosol acidity or aerosol chloride. Shipborne measurements off the LA coast did not observe elevated daytime Cl₂ most of the time (Riedel et al., 2012). Aerosol information was not reported in that paper. We would think that chloride from sea salt should be abundant in that environment and nitrate should be sufficient during periods of off-shore winds. Their E-AIM model calculated that submicrometer aerosol acidity (pH) was larger than 4. According to the pH-dependency of Cl₂ production in our experiments, the Cl₂ production is expected to be very slow under such conditions, which could account for the absence of Cl₂ in the daytime in their study. For the oil-exploration impacted remote Uintah basin, a small amount of ClNO₂ was reported (Edwards et al., 2014), and the author attributed the low ClNO₂ to lacking aerosol-phase chloride, which may also explain the non-existence of Cl₂ in that study.

On the other hand, elevated daytime concentrations of Cl₂ have been observed in the northern China Plain such as Beijing megacity in summer (Qiu et al., 2019) and

semi-rural Wangdu (Liu et al., 2017), suburban Nanjing in spring (Xia et al., 2020), Manchester city (Priestley et al., 2018), although the concentrations are not nearly as high as in our study. We believe that the co-existence of aerosol-phase chloride (in both coastal and inland areas in China), high levels of nitrate (due to high NO_x emission), and high acidity could produce a great deal of the daytime Cl₂. Because there are still limited observations of Cl₂ and the influencing chemical parameters, we call for more such studies in other geographical locations. We have added the following discussion in the revised manuscript to explain the lack of daytime Cl₂ in previous studies:

Lines 300-310, *“The above results indicate the importance of the coexistence of the three key factors in Cl₂ production, namely, nitrate, chloride, and acidity in the aerosol particles. Our proposed Cl₂ production mechanism could qualitatively explain the lack of daytime Cl₂ in previous studies that reported higher aerosol acidity or lower aerosol chloride content. During shipborne measurements off the coast of Los Angeles, elevated Cl₂ concentrations were observed mostly at night and in isolated industrial plumes¹⁵. Their E-AIM model calculated sub-micrometer aerosol pH was > 4. According to the pH-dependency of Cl₂ production in our experiments, the Cl₂ production is expected to be very slow under such conditions, which could explain the absence of daytime Cl₂ in their study. Very low levels of ClNO₂ (and lack of Cl₂) were reported in the oil-exploration impacted Uintah basin⁴⁴, and the author attributed the low ClNO₂ to lacking aerosol-phase chloride at this remote inland site, which would also explain the absence of Cl₂ in that study.”*

2. What does the chemistry and associated box-modeling imply for levels of gas-phase HCl? I would imagine they have to be quite high in order to maintain a continuous source of soluble chloride to the particles. Was HCl measured, and what does the model require to make the chemistry work?

Response: We did not measure gas-phase HCl. As our site is near the coast, sea salt is the main source of aerosol chloride, some of which can be displayed by HNO₃ to produce HCl which could be taken up again into the aerosol phase. The E-AIM estimated an average ambient gas-phase HCl concentration of 0.96 ppbv (standard deviation: 0.52 ppbv) and an average equivalent Cl⁻ molarity in PM_{2.5} of 0.10 mol L⁻¹ (standard deviation: 0.19 mol L⁻¹) during the field study. Therefore, chloride is sufficiently abundant to sustain the Cl₂ production. Our box model is constrained by the observed Cl₂ and does not require high levels of HCl. We have added the equivalent Cl⁻ molarity and HCl information in the caption of Supplementary Fig. 10 on E-AIM model. We have now indicated the calculated gas phase HCl abundance.

Lines 86-88, *“Supplementary Fig. 10. ... The E-AIM estimated average equivalent Cl⁻ molarity in PM_{2.5} was 0.10 mol L⁻¹ (standard deviation: 0.19 mol L⁻¹) and gas-phase HCl concentration was 0.96 ppbv (standard deviation: 0.52 ppbv).”*

3. The reaction scheme for N and Cl species seems incomplete as presented, since we know that Cl₂ reaction on chloride-containing substrates can be used as a facile source of ClNO₂ (Thaler et al., 2011). This reaction is undoubtedly pH dependent, as it appears to go in the reverse direction below pH 2 (Roberts et al., 2008). Low pH conditions will

of course liberate NO_2^- as HONO, and Cl^- as HCl, so could it be that there is a lower limit on the pH conditions at which this chemistry is happening? I would like to see more discussion of this.

Response: Thanks for noting the conversion reaction between Cl_2 and ClNO_2 . The referee mentioned reaction is $\text{Cl}_2 + \text{NO}_2^- \rightleftharpoons \text{ClNO}_2 + \text{Cl}^-$ (Thaler et al., 2011). At pH less than 2, the reverse reaction is favored, and produces Cl_2 .

Our reaction scheme lists the previously recognized major daytime Cl_2 producing reactions. As ClNO_2 is typically at a very low-level during daytime (which is the case in our study), we do not think that ClNO_2 is the source for Cl_2 during the daytime. Thus, we did not list that reaction in the daytime reaction scheme shown in our paper.

We agree that there could be a lower pH limit for Cl_2 production. As pointed out by the referee, if pH were too low, Cl^- would be converted to HCl and NO_2^- to HONO, the former would reduce Cl_2 production and the latter would promote Cl_2 formation. E-AIM model estimated $\text{PM}_{2.5}$ pH at our study site ranges from 0.5 to 3.0 (average: 1.6) when very high levels of Cl_2 were observed. A recent review of global aerosol acidity (Pye et al., 2020) indicates that about 90% of fine-mode aerosol have pH larger than 0.5 and 80% above 1.6. We suggest that the lower pH limit would not be reached for typical tropospheric aerosols. That is, aerosol pH with below 3.0 should promote Cl_2 production and much more acidic aerosols, which would lose Cl^- , are not prevalent. Bottom line: Yes, there is a sweet-spot for pH that enables high Cl_2 production.

We have added this discussion in the revised manuscript, shown as below:

Lines 311-321, *“Our result has shown that pH below 3.3 is a necessary condition for nitrate-induced Cl_2 production. There could be a lower limit of pH for this production pathway. If the pH were too low, Cl^- would be converted to HCl and NO_2^- to HONO, the former would reduce Cl_2 production while the latter would promote Cl_2 formation. Accordingly, there could be a window of suitable pH conditions for the described way of Cl_2 formation to efficiently take place. The estimated average pH for $\text{PM}_{2.5}$ at our study site is 1.6 (range: 0.5-3.0), and a very large production of Cl_2 was observed under such acidic conditions. A recent review of global aerosol acidity⁴⁵ indicates that about 90% of fine-mode aerosol has a pH larger than 0.5, with 80% above 1.6. We suggest that the lower pH limit at which Cl_2 production would decrease may not be reached for typical tropospheric aerosols. That is, aerosol pH smaller than 3.0 should promote Cl_2 production under most tropospheric conditions.”*

In addition, we have replaced the original simplified R(5) with detailed reaction sequence in the revised manuscript, as shown below (Lines 117-121):

4. There are several paragraphs where the authors stray off into making concluding remarks that seem out of place, and belong in the concluding section which is named “Impact on Atmospheric Chemistry and Implications”. I will point those out below.

Response: We agree. The related paragraphs have been amended and merged into the concluding section. Please see below for a detailed response.

Specific Comments:

5. Line 57, ‘VOCs’ should be ‘VOC’

Response: Thanks. Referee 3 suggest that methane is not part of VOC defined in air quality regulations. We think it is more appropriate to change it to “hydrocarbon” here. So the sentence has been modified as below:

Lines 51-52, *“Cl reacts rapidly with methane, the most abundant hydrocarbon and the second-most important greenhouse gas emitted into the atmosphere^{8,9}.”*

6. Line 107-110. The chemistry as presented in R3-R5 is at least second-order in Cl⁻, (maybe 3rd-order?). Was that factored into the extrapolation?

Response: As seen above, we have now detailed the different steps involved in Cl₂ production. The extrapolation method is the same as that of Oum et al. 1998 who extrapolated the Cl₂ production to typical mid-Atlantic ozone and solar radiation. It was assumed that the Cl₂ production is proportional to the level of ozone and solar radiation and Cl⁻ is sufficient in that study. We clarified the text as follows:

Lines 126-130, *“This production rate was extrapolated to typical mid-Atlantic conditions, assuming that the Cl₂ production was proportional to the level of ozone and solar radiation and Cl⁻ availability was sufficient. These conditions explained the observed Cl₂ at a coastal site in Long Island, New York²⁷. If we extrapolate their production rate, with the same assumptions, to our ambient conditions...”*

7. Line 156-161. It seems to me that one key quantity here is the equivalent Cl⁻ molarity in the particles, assuming they are droplets. What were those numbers?

Response: The E-AIM estimated an average equivalent Cl⁻ molarity in PM_{2.5} of 0.10 mol L⁻¹ (standard deviation: 0.19 mol L⁻¹). This information has been added in the caption of Fig S10.

Lines 86-88, *“Supplementary Fig. 10. ... The E-AIM estimated average equivalent Cl⁻ molarity in PM_{2.5} was 0.10 mol L⁻¹ (standard deviation: 0.19 mol L⁻¹) and gas-phase HCl concentration was 0.96 ppbv (standard deviation: 0.52 ppbv).”*

8. Fig. 1. I can’t see the labels on the bottom of panels C&D.

Response: We enlarged the labels, and please see below.

Lines 788-789, “Fig. 1. Ambient observations at Hok Tsui, Hong Kong, from 31 August to 9 October 2018....”

9. Line 203 and Figure S8. This was really confusing as stated. Figure S8A shows increasing Cl_2 , so when the authors said “no increase in Cl_2 signals”, it took me a while to figure out they mean no relative increase with O_3 added compared to no O_3 . The language needs to be clarified here. Also, the text here says pH 2.9 and the Figure says 3.9, so which is it?

Response: We apologize for not stating this clearly. Fig. 8A showed one case (pH = 1.9) of low pH condition from 1.9 to 2.9, and Fig. 8B showed one case of relatively high pH condition from 3.3 to 6.8 (pH = 3.9).

We have changed the text as:

Lines 196-198, “We also did not detect Cl_2 when the zero air containing various O_3 mixing ratios (150, 250, and 500 ppbv) flowed over the illuminated solution of 1 M sodium chloride.”

Lines 209-214, “We also investigated the influence of ozone on Cl_2 production. There was no relative increase in the Cl_2 signals when zero air containing differing O_3 mixing ratios (150, 250, and 500 ppbv) flowed over the illuminated chloride-nitrate solutions with a pH of 1.9 to 2.9, compared to the no O_3 cases (Supplementary Fig. 8A for pH=1.9). And the Cl_2 level also did not increase when the added O_3 increased from 150 ppbv to 500 ppbv with a pH of 3.3 to 6.8 (Supplementary Fig. 8B for pH=3.9).”

10. Line 242-244. Did you see an increase in $ClNO_2$ anytime when adding NO_2^- ?

Response: We did observe $ClNO_2$ when NO_2^- was added (see below figure). This phenomenon had been previously observed and was attributed to $Cl_2 + NO_2^- \rightarrow ClNO_2 + Cl^-$ (Thaler et al., 2011).

Figure R1. Formation mechanism experimental results on solutions. Time series of 1-min average CINO₂ and NO_x. The liquid solution sample (pH=1.95) was illuminated at t=0. The solid black line shows the time at which 10ul OH scavenger, TBA, was added, the black dashed line indicates the time at which 10ul DI water was added, the black point line indicates the time at which 10ul nitrite was added, and the red point line indicates the time at which the xenon lamp was turned off. The short-term signal increase of CINO₂ was due to the higher concentration of CINO₂ in the room when we opened the chamber and added the TBA, DI water, and nitrite.

11. Lines 264-276. Much of this material belongs in the concluding section.

Response: Thanks for the suggestion, we have condensed this part and moved one implication to the concluding section.

Lines 374-376, “...an interesting coupling between condensed phase oxidation resulting in the formation of its gaseous counterpart, which is expected to have important implications on atmospheric chemistry and production of secondary air pollutants. ...”

12. Line 295-308. This paragraph belongs in the concluding section.

Response: Thanks for the suggestion. We have moved and merged them in the concluding section.

Lines 350-372, “In summary, a limited number of prior studies have indicated the presence and important role of daytime Cl₂ in the photochemistry of the lower troposphere in polluted regions. However, the exact source or production mechanism remained uncertain, which has hindered the reproduction of the daytime Cl₂ in current state-of-the-art global and regional chemistry transport models. In the present study, we observed the highest Cl₂ concentration ever reported to date at a polluted coastal site, implying that the Cl₂ could exist in more places and at higher concentrations than previously thought, in view of the limited Cl₂ observations to date. Combining laboratory and field measurements, we show that Cl₂ can be produced from photolysis of aerosol containing nitrate, chloride, and high acidity, and demonstrate that this

mechanism can explain a great fraction of the observed daytime Cl₂ at our measurement site. Our result indicates the critical role of aerosol acidity (pH <3.3) in promoting Cl₂ production. The same mechanism may occur in other parts of the world impacted by anthropogenic pollution. Despite a significant reduction in the emissions of acid precursors like sulfur dioxide (SO₂) and nitrogen oxides (NO_x), highly acidic aerosols are still present in some areas/seasons in Asia, North America, and Europe⁴⁵, and nitrate aerosols are also abundant in world's urban and industrial regions^{46,47}. Previous studies have also indicated the ubiquity of aerosol chloride in continental as well as maritime environments^{3,30,48,49}. We, therefore, anticipate that the Cl₂ production operates in some places or times where/when sufficiently high levels of acidity, nitrate, and chloride co-exist. Our recent study in northern China also shows that nitrate photolysis could activate chloride and bromide in coal-burning aerosol, which exerted a large impact on winter oxidation chemistry⁶. We note that elevated nitrate and the other acidic aerosol (sulfate) have been observed during the Arctic haze events^{50,51}. It would be of great interest to investigate whether the nitrate photolysis mechanism would contribute to the liberation of inert chlorine in the polar troposphere.”

13. Line 350. What was the ionization chemistry of the HR-ToF-MS? I would assume proton-transfer-reaction (PTR), but that needs to be specified here.

Response: The HR-ToF-CIMS ionizes the target gases using iodide (I⁻) as the reagent ion, not proton ion. Cl₂ and thirteen gas-phase C₁-C₆ ClOVOCs were detected, with 1-chloro-3-methyl-3-butene-2-one (CMBO, C₅H₆ClO) as the most dominant organochloride. Cl₂ and CMBO were detected as iodide adducts (ICl₂⁻ and IC₅H₆ClO⁻, respectively) after ion-molecule reactions: I⁻ + Cl₂ → ICl₂⁻, I⁻ + C₅H₆ClO → IC₅H₆ClO⁻. Other species were measured with the similar ionization chemistry. We have added below information in the caption of Supplementary Fig. 13.

Lines 110-116, “Briefly, the HR-ToF-CIMS adopts chemical reactions to ionize the target gases using iodide (I⁻) as the reagent ion. Along with Cl₂, thirteen gas-phase C₁-C₆ ClOVOCs were detected, with 1-chloro-3-methyl-3-butene-2-one (CMBO, C₅H₆ClO) as the most dominant organochloride. Cl₂ and CMBO were detected as iodide adducts (ICl₂⁻ and IC₅H₆ClO⁻, respectively) after ion-molecule reactions: I⁻ + Cl₂ → ICl₂⁻, I⁻ + C₅H₆ClO → IC₅H₆ClO⁻. Other species were measured with the similar ionization chemistry.”

14. Line 441. Did you observe any differences in signals or backgrounds after washing the inlet?

Response: In the field, we washed the inlet when ambient Cl₂ signals were low, and it took about 2 hours to wash and dry the inlet. Because of the time lag and low Cl₂ concentrations, we could not observe the effect of inlet washing on ambient Cl₂ signals. However, the post-campaign lab test indicated much less loss in a new tubing than that in a field used tubing. Thus, we believe that inlet washing in the field was effective to reduce possible sampling artifacts.

15. Line 462. I can't tell what is meant by “scrubbing the ambient air”?

Response: We are sorry for this incorrect description. It has been changed.

Lines 436-438, “...*(i) potential inlet artifact arising from heterogeneous reactions of ambient O₃ and N₂O₅ were tested by adding O₃ and N₂O₅ into the ambient air in the intake to the measurement system.*”

16. Line 500. Don't you also need gas phase HNO₃ and HCl for the model.

Response: E-AIM model does not require the HNO₃ and HCl concentrations as input information.

17. Line 543. The number has too many significant figures.

Response: Thanks for pointing out this, it has been changed to “ 5.1×10^5 ”.

References:

1. Riedel, T. P. et al. Nitryl Chloride and Molecular Chlorine in the Coastal Marine Boundary Layer. *Environmental Science & Technology* 46, 10463-10470, doi:10.1021/es204632r (2012).
2. Edwards, P. M. et al. High winter ozone pollution from carbonyl photolysis in an oil and gas basin. *Nature* 514, 351-354, doi:10.1038/nature13767 (2014).
3. Qiu, X. et al. Modeling the impact of heterogeneous reactions of chlorine on summertime nitrate formation in Beijing, China. *Atmos. Chem. Phys.* 19, 6737-6747, doi:10.5194/acp-19-6737-2019 (2019).
4. Liu, X. et al. High levels of daytime molecular chlorine and nitryl chloride at a rural site on the North China Plain. *Environmental science & technology* 51, 9588, doi:10.1021/acs.est.7b03039 (2017).
5. Xia, M. et al. Significant production of ClNO₂ and possible source of Cl₂ from N₂O₅ uptake at a suburban site in eastern China. *Atmos. Chem. Phys.* 20, 6147-6158, doi:10.5194/acp-20-6147-2020 (2020).
6. Priestley, M. et al. Observations of organic and inorganic chlorinated compounds and their contribution to chlorine radical concentrations in an urban environment in northern Europe during the wintertime. *Atmospheric Chemistry Physics* 18, 13481-13493 (2018).
7. Oum, K. W., Lakin, M., DeHaan, D. O., Brauers, T. & Finlayson-Pitts, B. J. Formation of molecular chlorine from the photolysis of ozone and aqueous sea-salt particles. *Science* 279, 74-76 (1998).
8. Thaler, R. D., Mielke, L. H. & Osthoff, H. D. Quantification of Nitryl Chloride at Part Per Trillion Mixing Ratios by Thermal Dissociation Cavity Ring-Down Spectroscopy. *Analytical Chemistry* 83, 2761-2766, doi:10.1021/ac200055z (2011).

Response to Reviewer Comments #2

Reviewer #2 (Remarks to the Author):

1. General comments: This manuscript reports a combination of field measurements, laboratory experiments and modeling that document the photochemical production of Cl_2 in air during the day at mid-latitudes and the impacts on VOC oxidation etc. The conclusion is that nitrate ion photochemistry, which is well known to generate OH under acidic conditions, is responsible via OH production followed by its oxidation of chloride ions. The manuscript is well-written and it is comprehensive in its approach.

The major pieces are well-known: (1) nitrate photochemistry generates OH; (2) OH oxidizes halide ions such as chloride to form Cl_2 ; (3) Cl_2 photolyzes to generate Cl atoms that oxidize VOC, leading to chemistry similar to that initiated by OH radicals in the gas phase; (4) if the production is sufficiently fast compared to the rate of Cl_2 photolysis, measurable levels of Cl_2 build up during the day. With that background, the results reported are not surprising, but it is a nice combination of field, lab and modeling that quantifies these processes for this particular location and rules out other contributing processes as being important. The appropriateness of this paper for Nature rather than a specialty journal could be strengthened if the authors could better articulate the unique and transformative aspects of the work.

Response: We thank the referee for a nice summary of our work. The uniqueness and transformative impact of our study are highlighted as follows.

The source and production mechanism for daytime Cl_2 outside the polar regions are uncertain because the previously proposed ozone mechanism (via production of OH followed by the oxidation of Cl⁻) is too slow to account for the elevated levels of Cl_2 observed during daytime, including our site. Although nitrate photolysis is known to produce OH, previous lab studies of illuminating nitrate with sea salt or mixtures of chloride and bromide did not produce detectable amounts of Cl_2 (George & Anastasio, 2007, Abbatt et al., 2010, Richards & Finlayson-Pitts, 2012). Our study, for the first time, demonstrates that significant amount of Cl_2 can be produced by illuminating nitrate-NaCl solution at pH below 3.3 and ambient aerosol samples. We further demonstrate that the confluence of nitrate photolysis, high acidity, and chloride abundances under condition typical of semi-polluted coastal environment could produce sufficient Cl_2 to quantitatively explain the daytime Cl_2 observations at a particular location. This combination, if not present, does not lead to Cl_2 production and thus explain the absence of Cl_2 in some measurement. Our result offers a new direction for developing the Cl_2 scheme to predict the Cl_2 impact on oxidation chemistry for air quality models that currently lack such capability. Our results indicate an important coupling of nitrogen- HO_x -halogen chemistry and suggest a previously unrecognized benefit of widely adopted anti-pollution measures for controlling SO_2 and NO_x . That is, reducing SO_2 and NO_x emissions not only reduces their adverse impacts on human health and welfare directly or through the formation of acid deposition and particles, but also decreases particle acidity and nitrate content, both of which would slow the Cl_2 production and its contribution to secondary pollution.

Moreover, there are just a few Cl_2 observations to date outside the polar regions, which limits our ability to assess the Cl_2 impact in other parts of the world. Our study

observed the highest Cl₂ concentration ever reported to date, revealing that Cl₂ could exist in more locations and at higher concentrations than previously thought, which calls for the atmospheric chemistry community to pay more attention to the role of Cl₂ in the polluted troposphere. The above uniqueness and broader impact of our study justifies its publication in Nature Communications.

We have articulated unique and transformative aspects of our study by adding texts in various places in the revised manuscript.

Instruction section:

Lines 67-69, *“Finally, as there were just only handfuls of Cl₂ observations outside the polar regions to date^{13-17,19-21}, our ability to assess the Cl₂ and Cl impact in different parts of the world is still very limited.”*

Concluding section:

Lines 353-361, *“.... In the present study, we observed the highest Cl₂ concentration ever reported to date at a polluted coastal site, implying that the Cl₂ could exist in more places and at higher concentrations than previously thought, in view of the limited Cl₂ observations to date. Combining laboratory and field measurements, we show that Cl₂ can be produced from photolysis of aerosol containing nitrate, chloride, and high acidity, and demonstrate that this mechanism can explain a great fraction of the observed daytime Cl₂ at our measurement site. Our result indicates the critical role of aerosol acidity (pH <3.3) in promoting Cl₂ production. The same mechanism may occur in other parts of the world impacted by anthropogenic pollution.”*

Lines 373-383, *“Our findings have indicated a previously unrecognized role of the reactive nitrogen cycle in both halogen and HO_x chemistry and, at the same time, an interesting coupling between condensed phase oxidation resulting in the formation of its gaseous counterpart, which is expected to have important implications on atmospheric chemistry and production of secondary air pollutants. These findings suggest a new direction for developing the Cl₂ scheme to predict its impact on oxidation chemistry for air quality models that currently do not include such chemistry. Moreover, our results suggest an additional benefit of widely adopted anti-pollution measures to control SO₂ and NO_x. That is, reducing SO₂ and NO_x emissions not only alleviates their adverse impacts on health and welfare directly and through the formation of acid deposition and particles, but also decreases particle acidity and nitrate, both of which would slow the Cl₂ production and its promoting consequences to secondary pollution production, for example, surface ozone.”*

Lines 383-387, *“We call for more investigations of the roles of halogen chemistry in the polluted troposphere and suggest some research that would place Cl₂ (and other halogens) production and atmospheric impact on a firmer footing. They include more atmospheric measurements of Cl₂ together with aerosol acidity (or its proxies), nitrate, and other parameters in diverse geographical areas, ...”*

Minor comments:

2. There is a long history regarding halogen chemistry in the troposphere. While the authors cannot cite every paper, perhaps the first specific measurement of tropospheric Cl₂ (Spicer et al, 1998) and some of the recent papers from the Pratt, Thornton, Bertram etc groups might be useful to readers.

Response: Thanks for suggesting the references. Papers from Spicer, Thornton, and Bertram were cited in the original manuscript. We have added one new reference from Pratt's group (McNamara et al., 2019).

Lines 55-58, *“Previously, Cl₂ has been measured in the lower troposphere in locations such as at the Arctic surface^{11,12}, the marine boundary layer¹³⁻¹⁵, and continental sites^{16,17}. Cl₂ was found to typically peak during nighttime, but elevated levels (17-450 ppt) have also been observed during daytime^{6,11,12,18-21}.”*

3. There is a lot of literature on nitrate ion photochemistry with sea salt, or mixtures of chloride and bromide that mimic sea salt, e.g. Anastasio 2007, Abbatt 2010, Finlayson-Pitts, 2012. It appears that Br⁻ is oxidized first but when bromide is depleted, Cl₂ is observed, which is similar to what is reported here. Citing some of these studies might be helpful.

Response: Thanks for suggesting these references. These laboratory studies were conducted under similar but not identical conditions (i.e. They used a UV lamp and sea salt or mixtures of chloride and bromide). They showed that reactive bromine gases (Br₂ and BrCl) were produced over acid-doped nitrate-halide solution (liquid or frozen), but Cl₂ was not observed, unlike our experiment. We also observed production of Br₂ and BrCl (see below figure). As our paper is about Cl₂, we prefer not to further discuss Br₂ and BrCl in detail here, but have added the information in the revised version.

Lines 205-209, *“Previous laboratory studies of halogen production under similar but not identical conditions (i.e. light source and reaction medium) indicated that reactive bromine gases (Br₂ and BrCl) were produced over acid-doped nitrate-halide solution (liquid or frozen) under UV light (~310 nm)³¹⁻³³, but Cl₂ was not observed unlike our experiment. Note that in our study, Br₂ and BrCl were also produced together with Cl₂.”*

Figure R2. Experimental results on solutions in the dynamic chamber. Time series of 1-min average Cl_2 , BrCl , Br_2 , HONO , and NO_x with the acidic liquid solution samples of $\text{pH}=1.89$. Sample was illuminated at $t=0$. The solid red line shows the time at which 300-800 nm filter was used, the red dashed line indicates the time at which AM1.5 filter was used, and the red point line indicates the time at which the xenon lamp was turned off. The left inset: enlarged figures during the time -60 ~ 60 min. The right inset: the scan mass spectra from 196 amu to 200 amu during the experiments at $t = 387$ min.

4. It was not clear to this reviewer why a stronger correlation occurs when particle surface area is included in the analysis. Their model results indicate that interface reactions R3-R5 cannot be important. Did they look at a correlation such as solar radiation $\times \text{Cl}^- \times \text{NO}_3^-$, i.e. the major active species without the surface area? If it is nitrate photochemistry at the interface, is there likely to be enough nitrate at the interface? Could preferential occupation of ions at the interface (e.g. Richards-Henderson 2013) play a role?

Response: We think that production of Cl_2 by nitrate photolysis mainly occurs at the surface, which is supported by the lab results of P_{Cl_2} increasing with the surface area (see the response to a related comment later).

The P_{Cl_2} showed only a weak correlation with the following factor (solar $\times \text{Cl}^- \times \text{NO}_3^-$), as shown below, supporting the idea that the surface area plays an important role in Cl_2 production. This is also supported by some kinetic modelling (Knipping et al., 2000) and laboratory studies (Richards-Henderson et al., 2013) which were suggesting preferential occupation of nitrate ions at the interface. We have added the below text:

Lines 175-176, “... (Note that the correlation was largely decreased ($R^2=0.39$) if the surface area density was excluded.)”

Lines 264-266, “Previous kinetic modelling²⁸ and laboratory studies⁴³ indicated preferential occupation of nitrate ions at the interface, which can facilitate fast surface reactions.”

Figure R3. The scatter plot of the production rate of Cl₂ (P_{Cl₂}) and the product of the solar actinic flux, the chloride concentration, and nitrate concentration in PM₁₀ (ug/m³) from 08:00 to 18:00 in the continental air mass.

5. Lines 191-199: the statement that “We did not detect Cl₂ when the zero air containing various O₃ mixing ratios.... pH from 3.3 to 6.8. One example is shown in Fig. S8”. But in Fig S8B, Cl₂ is observed at pH 3.9 as O₃ is increased. Clarification?

Response: We apologize for not stating this clearly. Fig. 8A showed one case (pH = 1.9) of low pH condition from 1.9 to 2.9, and Fig. 8B showed one case of relatively high pH condition from 3.3 to 6.8 (pH = 3.9).

We have changed the text as:

Lines 196-198, “We also did not detect Cl₂ when the zero air containing various O₃ mixing ratios (150, 250, and 500 ppbv) flowed over the illuminated solution of 1 M sodium chloride.”

Lines 209-214, “We also investigated the influence of ozone on Cl₂ production. There was no relative increase in the Cl₂ signals when zero air containing differing O₃ mixing ratios (150, 250, and 500 ppbv) flowed over the illuminated chloride-nitrate solutions with a pH of 1.9 to 2.9, compared to the no O₃ cases (Supplementary Fig. 8A for pH=1.9). And the Cl₂ level also did not increase when the added O₃ increased from 150 ppbv to 500 ppbv with a pH of 3.3 to 6.8 (Supplementary Fig. 8B for pH=3.9).”

6. In the lab experiments, more Cl₂ is observed when the 4 mL of solution is split into 4 x 1 mL samples, and it is argued that this is due to an increase in surface area. However, could it be due to increased diffusion into the gas phase from the increased surface area?

Response: Thanks for suggesting this explanation. We have added it in the revised manuscript.

Lines 261-266, *“We also investigated the effect of surface area on Cl₂ production. In the laboratory experiments, more Cl₂ was observed when 4 mL of the nitrate-NaCl solution was split into 4 × 1 mL samples (Supplementary Fig. 9). This may be explained by increased Cl₂ production and diffusion into the gas phase from the increased surface area. Previous kinetic modelling²⁸ and laboratory studies⁴³ indicated preferential occupation of nitrate ions at the interface, which can facilitate fast surface reactions.”*

7. Lines 295-308: Regarding the generality of the chemistry, the authors may want to include the possibility of chloride in road salt in winter climates as contributors (Ariya et al, 2020; papers by Pratt, Osthoff groups).

Response: Thanks for the suggestion, we have added this reference (McNamara et al., 2020) there.

Lines 364-365, *“Previous studies have also indicated the ubiquity of aerosol chloride in continental as well as maritime environments^{3,30,48,49}.”*

8. Figure S13: Some references to the identification of CMBO as a chlorine atom tracer (e.g. Nordmeyer et al, 1997; Tanaka et al, 2003) might be useful here.

Response: Thanks. The references have been added.

Lines 116-117, *“Supplementary Fig. 13.... CMBO is the chlorine oxidation product of isoprene, which makes this CLOVOC a unique tracer of chlorine-biogenic chemistry^{2,3}.”*

References:

1. McNamara, S. M. et al. Springtime Nitrogen Oxide-Influenced Chlorine Chemistry in the Coastal Arctic. *Environmental Science & Technology* 53, 8057-8067, doi:10.1021/acs.est.9b01797 (2019).
2. George, I. J. & Anastasio, C. Release of gaseous bromine from the photolysis of nitrate and hydrogen peroxide in simulated sea-salt solutions. *Atmospheric Environment* 41, 543-553, doi:https://doi.org/10.1016/j.atmosenv.2006.08.022 (2007).
3. Abbatt, J. et al. Release of Gas-Phase Halogens by Photolytic Generation of OH in Frozen Halide–Nitrate Solutions: An Active Halogen Formation Mechanism? *The Journal of Physical Chemistry A* 114, 6527-6533, doi:10.1021/jp102072t (2010).
4. Richards, N. K. & Finlayson-Pitts, B. J. Production of Gas Phase NO₂ and Halogens from the Photochemical Oxidation of Aqueous Mixtures of Sea Salt and Nitrate Ions at Room Temperature. *Environmental Science & Technology* 46, 10447-10454, doi:10.1021/es300607c (2012).

5. Knipping, E. et al. Experiments and simulations of ion-enhanced interfacial chemistry on aqueous NaCl aerosols. *Science* 288, 301-306 (2000).
6. Richards-Henderson, N. K. et al. Production of gas phase NO₂ and halogens from the photolysis of thin water films containing nitrate, chloride and bromide ions at room temperature. *Physical Chemistry Chemical Physics* 15, 17636-17646, doi:10.1039/C3CP52956H (2013).
7. McNamara, S. M. et al. Observation of road salt aerosol driving inland wintertime atmospheric chlorine chemistry. *American Chemical Society Central Science* 6, 684-694, doi:10.1021/acscentsci.9b00994 (2020).

Response to Reviewer Comments #3

Reviewer #3 (Remarks to the Author):

1. General comments: The authors measured high levels of Cl₂ in a coastal area in Hong Kong, the formation of which cannot be explained by known/recognized processes of Cl₂ formation from particulate chloride. Further analysis of the ambient data and laboratory experiments is consistent with Cl₂ activation via nitrate photolysis under acidic conditions. This topic is important and relevant to the field of atmospheric chemistry and air pollution, the work is original, methodologies used are sound, and the paper is generally written well. My main concern and only major comment is that I find the main results and implications a bit overstated. More specifically, the authors convinced me that nitrate photolysis under acidic conditions can activate Cl₂. However, I don't necessarily agree that they have "unraveled the daytime source of molecular chlorine in the extra-polar atmosphere" (quoting their title). The source of Cl₂ in the atmosphere is subject of ongoing studies across the world, and there are likely other processes of heterogeneous Cl₂ formation (from particulate Cl₂) and probably also several sources of Cl₂ emissions that are not accounted for in the models. This would also be consistent with the authors' observation that nitrate photolysis explains 13-68% (i.e. not all) of the observed Cl₂ production. I would thus suggest and request that the authors acknowledge the (in my opinion, high) likelihood of other processes and sources contributing to the high observed Cl₂ concentrations. They may have found an important source/process, but not unraveled the source. I suggest these changes be made before publication of the manuscript.

Response: We appreciate the referee's comments. In the analysis, we have shown evidence of nitrate photolysis under high acidity is important in activating chloride. Our estimate indicates that airborne particles could account for 13-68% of observed Cl₂ production for aerosol pH in the range of 1.9-3.0. However, the lower limit contribution turns out to be too conservative as the E-AIM model calculated average aerosol pH is 1.5 at our site, so a more reasonable estimate should be 68%. We acknowledge uncertainties in the extrapolation of the lab results to ambient conditions and possibility of other source(s). The following amendments have been made in the revised manuscript:

The revised title "Unraveling a significant daytime source of molecular chlorine in the extra-polar atmosphere"

Also added the following in the text.

Lines 283-285, "The laboratory-determined Cl₂ production rates on liquid solutions at pH of 1.9 (similar to the average value of the ambient aerosol), and 1 mol L⁻¹ nitrate was 114 pptv s⁻¹ (shown at t = 520 min in Fig. 2)"

Lines 294-299, "We note that the above extrapolation is subject to uncertainty, including that from applying the Cl₂ production over the tested solution to the ambient aerosol, that from estimating aerosol pH with current aerosol thermodynamic models, and not accounting for competing reactions for OH and Cl, such as by organics. It is also possible that other unidentified source(s) may contribute to part of the observed daytime Cl₂."

I have included a few more minor comments below.

2. General note: I likely did not catch all minor editorial errors/issues. Please re-review the manuscript for these possible errors.

Response: Thanks for this helpful suggestion. We have checked again.

3. Line 47: remove “a”

Response: Thanks for pointing this out, it has been corrected.

4. Lines 56-57: my understanding is that methane is not considered a VOC (e.g. the US EPA explicitly excludes it from its definition of VOCs)

Response: Thanks for this suggestion. We have changed it to “*hydrocarbon*”

Lines 51-52, “*Cl reacts rapidly with methane, the most abundant hydrocarbon and the second-most important greenhouse gas emitted into the atmosphere*^{8,9}.”

5. Line 244: not clear what is meant by “lit solution”

Response: Sorry for this unclear, we have revised it as: “*in the illuminated solution*”.

Lines 235-237, “*we added 10 μ l 0.1 M Tert-Butyl Alcohol⁴⁰ (TBA, a scavenger of OH radical with a rate constant of $(3.8-7.6) \times 10^8 \text{ M}^{-1} \text{ s}^{-1}$) in the illuminated solution (Fig. 2B).*”

Lines 251-252, “*When we added a very small amount of NO_2^- (10 μ l, 0.01 M) in the illuminated solution, ...*”

6. Line 264: suggest removing “the” in front of “zero air”

Response: Thanks for pointing this out, it has been corrected.

7. Line 308: please review that line – there seem to be some errors in the prepositions used that make the sentence difficult to understand.

Response: Thanks for pointing this out. We have modified the sentence to:

Line of 370-372, “*It would be of great interest to investigate whether the nitrate photolysis mechanism would contribute to the liberation of inert chlorine in the polar troposphere.*”

REVIEWERS' COMMENTS

Reviewer #1 (Remarks to the Author):

The authors have answered the comments and questions adequately, so the paper is acceptable to me.

Reviewer #2 (Remarks to the Author):

The authors have responded to my comments/suggestions and I have no further comments.

Reviewer #3 (Remarks to the Author):

The authors have addressed my concerns in this revised manuscript, and I recommend publication. In the updated title I suggest replacing "significant" with a different word, e.g. "substantial" or "important"